# Sampling Sketches for
# Concave Sublinear Functions of Frequencies

**Edith Cohen**
Google Research, CA
Tel Aviv University, Israel
edith@cohenwang.com

**Ofir Geri**
Stanford University, CA
ofirgeri@cs.stanford.edu

## Abstract

We consider massive distributed datasets that consist of elements modeled as key-value pairs and the task of computing statistics or aggregates where the contribution of each key is weighted by a function of its *frequency* (sum of values of its elements). This fundamental problem has a wealth of applications in data analytics and machine learning, in particular, with *concave sublinear* functions of the frequencies that mitigate the disproportionate effect of keys with high frequency. The family of concave sublinear functions includes low frequency moments ($p \leq 1$), capping, logarithms, and their compositions. A common approach is to sample keys, ideally, proportionally to their contributions and estimate statistics from the sample. A simple but costly way to do this is by aggregating the data to produce a table of keys and their frequencies, apply our function to the frequency values, and then apply a weighted sampling scheme. Our main contribution is the design of composable sampling sketches that can be tailored to any *concave sublinear* function of the frequencies. Our sketch structure size is very close to the desired sample size and our samples provide statistical guarantees on the estimation quality that are very close to that of an ideal sample of the same size computed over aggregated data. Finally, we demonstrate experimentally the simplicity and effectiveness of our methods.

## 1 Introduction

We consider massive distributed datasets that consist of *elements* that are key-value pairs $e = (e.key, e.val)$ with $e.val > 0$. The elements are generated or stored on a large number of servers or devices. A key $x$ may repeat in multiple elements, and we define its *frequency* $\nu_x$ to be the sum of values of the elements with that key, i.e., $\nu_x := \sum_{e|e.key=x} e.val$. For example, the keys can be search queries, videos, terms, users, or tuples of entities (such as video co-watches or term co-occurrences) and each data element can correspond to an occurrence or an interaction involving this key: the search query was issued, the video was watched, or two terms co-occurred in a typed sentence. An instructive common special case is when all elements have the same value $1$ and the frequency $\nu_x$ of each key $x$ in the dataset is simply the number of elements with key $x$.

A common task is to compute statistics or aggregates, which are sums over key contributions. The contribution of each key $x$ is weighted by a function of its frequency $\nu_x$. One example of such sum aggregates are queries of domain statistics $\sum_{x \in H} \nu_x$ for some domain (subset of keys) $H$. The domains of interest are often overlapping and specified at query time. Sum aggregates also arise as components of a larger pipeline, such as the training of a machine learning model with parameters $\boldsymbol{\theta}$, labeled examples $x \in \mathcal{X}$ with frequencies $\nu_x$, and a loss objective of the form $\ell(\mathcal{X}; \boldsymbol{\theta}) = \sum_x f(\nu_x) L(x; \boldsymbol{\theta})$. The function $f$ that is applied to the frequencies can be any *concave sublinear* function. Concave sublinear functions, which we discuss further below, are used in

applications to mitigate the disproportionate effect of keys with very high frequencies. The training of the model typically involves repeated evaluation of the loss function (or of its gradient that also has a sum form) for different values of $\theta$. We would like to compute these aggregates on demand, without needing to go over the data many times.

When the number of keys is very large it is often helpful to compute a smaller random sample $S \subseteq \mathcal{X}$ of the keys from which aggregates can be efficiently estimated. In some applications, obtaining a sample can be the end goal. For example, when the aggregate is a gradient, we can use the sample itself as a stochastic gradient. To provide statistical guarantees on our estimate quality, the sampling needs to be *weighted* (*importance* sampling), with heavier keys sampled with higher probability, ideally, proportional to their contribution ($f(\nu_x)$). When the weights of the keys are known, there are classic sampling schemes that provide estimators with tight worst-case variance bounds [28, 13, 6, 10, 32, 33].

The datasets we consider here are presented in an *unaggregated* form: each key can appear multiple times in different locations. The focus of this work is designing composable sketch structures (formally defined below) that allow to compute a sample over unaggregated data with respect to the weights $f(\nu_x)$. One approach to compute a sample from unaggregated data is to first aggregate the data to produce a table of key-frequency pairs $(x, \nu_x)$, compute the weights $f(\nu_x)$, and apply a weighted sampling scheme. This aggregation can be performed using composable structures that are essentially a table with an entry for each distinct key that occurred in the data. The number of distinct keys, however, and hence the size of that sketch, can be huge. For our sampling application, we would hope to use sketches of size that is proportional to the desired sample size, which is generally much smaller than the number of unique keys, and still provide statistical guarantees on the estimate quality that are close to that of a weighted sample computed according to $f(\nu_x)$.

**Concave Sublinear Functions.** Typical datasets have a skewed frequency distribution, where a small fraction of the keys have very large frequencies and we can get better results or learn a better model of the data by suppressing their effect. The practice is to apply a *concave sublinear* function $f$ to the frequency, so that the importance weight of the key is $f(\nu_x)$ instead of simply its frequency $\nu_x$. This family of functions includes the frequency moments $\nu_x^p$ for $p \leq 1$, $\ln(1 + \nu_x)$, $\mathsf{cap}_T(\nu_x) = \min\{T, \nu_x\}$ for a fixed $T \geq 0$, their compositions, and more. A formal definition appears in Section 2.3.

Two hugely popular methods for producing word embeddings from word co-occurrences use this form of mitigation: word2vec [25] uses $f(\nu) = \nu^{0.5}$ and $f(\nu) = \nu^{0.75}$ for positive and negative examples, respectively, and GloVe [30] uses $f(\nu) = \min\{T, \nu^{0.75}\}$ to mitigate co-occurrence frequencies. When the data is highly distributed, for example, when it originates or resides at millions of mobile devices (as in federated learning [24]), it is useful to estimate the loss or compute a stochastic gradient update efficiently via a weighted sample.

The suppression of higher frequencies may also directly arise in applications. One example is campaign planning for online advertising, where the value of showing an ad to a user diminishes with the number of views. Platforms allow an advertiser to specify a cap value $T$ on the number of times the same ad can be presented to a user [19, 29]. In this case, the number of opportunities to display an ad to a user $x$ is a cap function of the frequency of the user $f(\nu_x) = \min\{T, \nu_x\}$, and the number for a segment of users $H$ is the statistics $\sum_{x \in H} f(\nu_x)$. When planning a campaign, we need to quickly estimate the statistics for different segments, and this can be done from a sample that ideally is weighted by $f(\nu_x)$.

**Our Contribution.** In this work, we design composable sketches that can be tailored to any concave sublinear function $f$, and allow us to compute a weighted sample over unaggregated data with respect to the weights $f(\nu_x)$. Using the sample, we will be able to compute unbiased estimators for the aggregates mentioned above. In order to compute the estimators, we need to make a second pass over the data: In the first pass, we compute the set of sampled keys, and in the second pass we compute their frequencies. Both passes can be done in a distributed manner.

A sketch $S(D)$ is a data structure that summarizes a set $D$ of data elements, so that the output of interest for $D$ (in our case, a sample of keys) can be recovered from the sketch $S(D)$. A sketch structure is composable if we can obtain a sketch $S(D_1 \cup D_2)$ of two sets of elements $D_1$ and $D_2$ from the sketches $S(D_1)$ and $S(D_2)$ of the sets. This property alone gives us full flexibility to

parallelize or distribute the computation. The size of the sketch determines the communication and storage needs of the computation.

We provide theoretical guarantees on the quality (variance) of the estimators. The baseline for our analysis is the bounds on the variance that are guaranteed by PPSWOR on aggregated data. PPSWOR [32, 33] is a sampling scheme with tight worst-case variance bounds. The estimators provided by our sketch have variance at most $4/((1 - \varepsilon)^2)$ times the variance bound for PPSWOR. The parameter $\varepsilon \leq 1/2$ mostly affects the run time of processing a data element, which grows near-linearly in $1/\varepsilon$. Thus, our sketch allows us to get approximately optimal guarantees on the variance while avoiding the costly aggregation of the data. We remark that these guarantees are for *soft* concave sublinear functions and the extension to any concave sublinear function incurs a factor of $(1 + 1/(e - 1))^2$ in the variance. The space required by our sketch significantly improves upon the previous methods (which all require aggregating the data). In particular, if the desired sample size is $k$, we show that the space required by the sketch at any given time is $O(k)$ in expectation and is well concentrated.

We complement our work with a small-scale experimental study. We use a simple implementation of our sampling sketch to study the actual performance in terms of estimate quality and sketch size. In particular, we show that the estimate quality is even better than the (already adequate) guarantees provided by our worst-case bounds. We additionally compare the estimate quality to that of two popular sampling schemes for aggregated data, PPSWOR [32, 33] and priority (sequential Poisson) sampling [28, 13]. In the experiments, we see that the estimate quality of our sketch is close to what achieved by PPSWOR and priority sampling, while our sketch uses much less space by eliminating the need for aggregation. This paper presents our sketch structures and states our results. The full version (including proofs and additional details) can be found in the supplementary material.

**Related Work.**   Composable weighted sampling schemes with tight worst-case variance for *aggregated* datasets (where keys are unique to elements) include priority (sequential Poisson) sampling [28, 13], VarOpt sampling [6, 10], and PPSWOR [32, 33]. We use PPSWOR as our base scheme because it extends to *unaggregated* datasets, where multiple elements can additively contribute to the frequency/weight of each key. A prolific line of research developed sketch structures for different tasks over streamed or distributed unaggregated data [26, 16, 1]. Composable sampling sketches for unaggregated datasets have the goal of meeting the quality of samples computed on aggregated frequencies while using a sketch structure that can only hold a final-sample-size number of distinct keys. Prior work includes a folklore sketch for distinct sampling ($f(\nu) = 1$ when $\nu > 0$) [22, 34], sum sampling ($f(\nu) = \nu$) [9, 18, 14, 11] based on PPSWOR, cap functions ($f(\nu) = \min\{T, \nu\}$) [7], and universal (multi-objective) samples with a logarithmic overhead that simultaneously support all concave sublinear $f$. In the current work we propose sampling sketches that can be tailored to any concave sublinear function and only have a small constant overhead. An important line of work uses random linear projections to estimate frequency statistics and to sample. In particular, $\ell_p$ sampling sketches [17, 27, 2, 21, 20] sample (roughly) according to $f(\nu) = \nu^p$. These sketches have higher overhead than sample-based sketches and are more limited in their application. Their advantage is that they can be used with super-linear (e.g., moments with $p \in (1, 2]$) functions of frequencies and can also support signed element values (the turnstile model). For the more basic problem of sketches that estimate frequency statistics over the full data, a characterization of sketchable frequency functions is provided in [5, 3]. Universal sketches for estimating $\ell_p$ norms of subsets were recently considered in [4]. A double logarithmic size sketch (extending [15] for distinct counting) that computes statistics over the entire dataset for all soft concave sublinear functions is provided in [8]. Our design builds on components of that sketch.

## 2   Preliminaries

Consider a set $D$ of data elements of the form $e = (e.key, e.val)$ where $e.val > 0$. We denote the set of possible keys by $\mathcal{X}$. For a key $z \in \mathcal{X}$, we let $\mathsf{Max}_D(z) := \max_{e \in D | e.key = z} e.val$ and $\mathsf{Sum}_D(z) := \sum_{e \in D | e.key = z} e.val$ denote the maximum value of a data element in $D$ with key $z$ and the sum of values of data elements in $D$ with key $z$, respectively. Each key $z \in \mathcal{X}$ that appears in $D$ is called *active*. If there is no element $e \in D$ with $e.key = z$, we say that $z$ is *inactive* and define $\mathsf{Max}_D(z) := 0$ and $\mathsf{Sum}_D(z) := 0$. When $D$ is clear from context, it is omitted. For a key $z$, we use the shorthand $\nu_z := \mathsf{Sum}_D(z)$ and refer to it as the frequency of $z$. The sum and the *max-distinct statistics* of $D$ are defined, respectively, as $\mathsf{Sum}_D := \sum_{e \in D} e.val$ and

$\mathsf{MxDistinct}_D := \sum_{z \in \mathcal{X}} \mathsf{Max}_D(z)$. For a function $f$, $f_D := \sum_{z \in \mathcal{X}} f(\mathsf{Sum}_D(z)) = \sum_{z \in \mathcal{X}} f(\nu_z)$ is the *$f$-frequency statistics* of $D$.

## 2.1 The Composable Bottom-$k$ Structure

In this work, we will use composable sketch structures in order to efficiently summarize streamed or distributed data elements. A composable sketch structure is specified by three operations: The initialization of an empty sketch structure $s$, the processing of a data element $e$ into a structure $s$, and the merging of two sketch structures $s_1$ and $s_2$. To sketch a stream of elements, we start with an empty structure and sequentially process data elements while storing only the sketch structure. The merge operation is useful with distributed or parallel computation and allows us to compute the sketch of a large set $D = \bigcup_i D_i$ of data elements by merging the sketches of the parts $D_i$. In particular, one of the main building blocks that we use is the bottom-$k$ structure [12], specified in Algorithm 1. The structure maintains $k$ data elements: For each key, consider only the element with that key that has the minimum value. Of these elements, the structure keeps the $k$ elements that have the lowest values.

---

**Algorithm 1:** Bottom-$k$ Sketch Structure

// **Initialize structure**
**Input:** the structure size $k$
$s.set \leftarrow \emptyset$ // `Set of` $\leq k$ `key-value pairs`
// **Process element**
**Input:** element $e = (e.key, e.val)$, a bottom-$k$ structure $s$
**if** $e.key \in s.set$ **then**
    replace the current value $v$ of $e.key$ in $s.set$ with $\min\{v, e.val\}$
**else**
    insert $(e.key, e.val)$ to $s.set$
    **if** $|s.set| = k + 1$ **then**
        Remove the element $e'$ with maximum value from $s.set$

// **Merge two bottom-$k$ structures**
**Input:** $s_1,s_2$ // `Bottom-k structures`
**Output:** $s$ // `Bottom-k structure`
$P \leftarrow s_1.set \cup s_2.set$
$s.set \leftarrow$ the (at most) $k$ elements of $P$ with lowest values (at most one element per key)

---

## 2.2 The PPSWOR Sampling Sketch

In this subsection, we describe a scheme to produce a sample of $k$ keys, where at each step the probability that a key is selected is proportional to its weight. That is, the sample we produce will be equivalent to performing the following $k$ steps. At each step we select one key and add it to the sample. At the first step, each key $x \in \mathcal{X}$ (with weight $w_x$) is selected with probability $w_x / \sum_y w_y$. At each subsequent step, we choose one of the remaining keys, again with probability proportional to its weight. This process is called probability proportional to size and without replacement (PPSWOR) sampling.

---

**Algorithm 2:** PPSWOR Sampling Sketch

// **Initialize structure**
**Input:** the sample size $k$
Initialize a bottom-$k$ structure $s.\mathsf{sample}$
 // `Algorithm 1`
// **Process element**
**Input:** element $e = (e.key, e.val)$, PPSWOR sample structure $s$
$v \sim \mathsf{Exp}[e.val]$
Process the element $(e.key, v)$ into the bottom-$k$ structure $s.\mathsf{sample}$
// **Merge two structures** $s_1, s_2$ `to obtain` $s$
$s.\mathsf{sample} \leftarrow$ Merge the bottom-$k$ structures $s_1.\mathsf{sample}$ and $s_2.\mathsf{sample}$

---

A classic method for PPSWOR sampling is the following scheme [32, 33]. For each key $x$ with weight $w_x$, we independently draw $\mathsf{seed}(x) \sim \mathsf{Exp}(w_x)$. The output sample will include the $k$ keys with smallest $\mathsf{seed}(x)$. This method together with a bottom-$k$ structure can be used to implement PPSWOR sampling over a set of data elements $D$ according to $\nu_x = \mathsf{Sum}_D(x)$. The sampling sketch is presented here as Algorithm 2. This sketch is due to [9] (based on [18, 14, 11]).

## 2.3 Concave Sublinear Functions

A function $f : [0, \infty) \to [0, \infty)$ is *soft concave sublinear* if for some $a(t) \geq 0$ it can be expressed as

$$f(\nu) = \mathcal{L}^c[a](\nu) := \int_0^\infty a(t)(1 - e^{-\nu t})dt . \tag{1}$$

$\mathcal{L}^c[a](\nu)$ is called the *complement Laplace transform* of $a$ at $\nu$. The sampling schemes we present in this work will be defined for *soft* concave sublinear functions of the frequencies. However, this will allow us to estimate well any function that is within a small multiplicative constant of a soft concave sublinear function. In particular, we can estimate *concave sublinear functions*. These functions can

**Algorithm 3:** Sampling Sketch Structure for $f$

// **Initialize empty structure** $s$
**Input:** $k$: Sample size, $\varepsilon$, $a(t) \geq 0$
Initialize $s.\mathsf{SumMax}$ // $\mathsf{SumMax}$ sketch of size $k$ (Algorithm 5)
Initialize $s.ppswor$ // PPSWOR sketch of size $k$ (Algorithm 2)
Initialize $s.sum \leftarrow 0$ // A sum of all the elements seen so far
Initialize $s.\gamma \leftarrow \infty$ // Threshold
Initialize $s.\mathsf{Sideline}$ // A composable max-heap/priority queue
// **Merge two structures** $s_1$ and $s_2$ to $s$ (with same $k, \varepsilon, a$ and same $h$ in $\mathsf{SumMax}$ sub-structures)
$s.sum \leftarrow s_1.sum + s_2.sum$
$s.\gamma \leftarrow \frac{2\varepsilon}{s.sum}$
$s.\mathsf{Sideline} \leftarrow$ merge $s1.\mathsf{Sideline}$ and $s2.\mathsf{Sideline}$ // merge priority queues.
$s.\mathsf{SumMax} \leftarrow$ merge $s_1.\mathsf{SumMax}$ and $s_2.\mathsf{SumMax}$ // Merge $\mathsf{SumMax}$ structures (Algorithm 5)
**while** $s.\mathsf{Sideline}$ *contains an element*
$\quad g = (g.key, g.val)$ *with* $g.val \geq s.\gamma$ **do**
$\quad\quad$ Remove $g$ from $s.\mathsf{Sideline}$
$\quad\quad$ **if** $\int_{g.val}^{\infty} a(t)dt > 0$ **then**
$\quad\quad\quad$ Process element $(g.key, \int_{g.val}^{\infty} a(t)dt)$ by
$\quad\quad\quad\quad s.\mathsf{SumMax}$

// **Process element**
**Input:** Element $e = (e.key, e.val)$, structure $s$
Process $e$ by $s.ppswor$
$s.sum \leftarrow s.sum + e.val$
$s.\gamma \leftarrow \frac{2\varepsilon}{s.sum}$
**foreach** $i \in [r]$ **do** // $r = k/\varepsilon$
$\quad y \sim \mathsf{Exp}[e.val]$ // exponentially distributed with parameter $e.val$
$\quad$ // Process in $\mathsf{Sideline}$
$\quad$ **if** *The key* $(e.key, i)$ *appears in* $s.\mathsf{Sideline}$ **then**
$\quad\quad$ Update the value of $(e.key, i)$ to be the minimum of $y$ and the current value
$\quad$ **else**
$\quad\quad$ Add the element $((e.key, i), y)$ to $s.\mathsf{Sideline}$

**while** $s.\mathsf{Sideline}$ *contains an element*
$\quad g = (g.key, g.val)$ *with* $g.val \geq s.\gamma$ **do**
$\quad\quad$ Remove $g$ from $s.\mathsf{Sideline}$
$\quad\quad$ **if** $\int_{g.val}^{\infty} a(t)dt > 0$ **then**
$\quad\quad\quad$ Process element $(g.key, \int_{g.val}^{\infty} a(t)dt)$ by
$\quad\quad\quad\quad s.\mathsf{SumMax}$

be expressed as

$$f(\nu) = \int_0^{\infty} a(t) \min\{1, \nu t\} dt \tag{2}$$

for $a(t) \geq 0$. The concave sublinear family includes all functions such that $f(0) = 0$, $f$ is monotonically non-decreasing, $\partial_+ f(0) < \infty$, and $\partial^2 f \leq 0$.

Any concave sublinear function $f$ can be approximated by a soft concave sublinear function as follows. Consider the corresponding soft concave sublinear function $\tilde{f}$ using the same coefficients $a(t)$. The function $\tilde{f}$ closely approximates $f$ pointwise [8]:

$$(1 - 1/e)f(\nu) \leq \tilde{f}(\nu) \leq f(\nu) .$$

Our weighted sample for $\tilde{f}$ will respectively approximate a weighted sample for $f$.

Consider a soft concave sublinear $f$ and a set of data elements $D$ with the respective frequency function $W : (0, \infty) \to \mathbb{N} \cup \{0\}$ (for every $\nu > 0$, $W(\nu)$ is the number of keys with frequency $\nu$ in $D$). The statistics $f_D = \sum_x f(\mathsf{Sum}_D(x)) = \sum_x f(\nu_x)$ can then be expressed as $f_D = \mathcal{L}^c[W][a]_0^{\infty}$ with the notation

$$\mathcal{L}^c[W][a]_{\gamma}^b := \int_{\gamma}^b a(t) \, \mathcal{L}^c[W](t) dt . \tag{3}$$

## 3 Sketch Overview

Given a set $D$ of elements $e = (e.key, e.val)$, we wish to maintain a sample of $k$ keys, that will be close to PPSWOR according to a soft concave sublinear function of their frequencies $f(\nu_x)$. At a high level, our sampling sketch is guided by the sketch for estimating the statistics $f_D$ due to Cohen [8].

Recall that a soft concave sublinear function $f$ can be represented as $f(w) = \mathcal{L}^c[a](w)_0^{\infty} = \int_0^{\infty} a(t)(1 - e^{-wt}) dt$ for $a(t) \geq 0$. Using this representation, we express $f(\nu_x)$ as a sum of two contributions for each key $x$:

$$f(\nu_x) = \mathcal{L}^c[a](\nu_x)_0^{\gamma} + \mathcal{L}^c[a](\nu_x)_{\gamma}^{\infty},$$

where $\gamma$ is a value we will set adaptively while processing the elements. Our sampling sketch is described in Algorithm 3. It maintains a separate sampling sketch for each set of contributions. In order to produce a sample from the sketch, these separate sketches need to be combined. Algorithm 4 describes how to produce a final sample from the sketch.

Running Algorithm 3 and then Algorithm 4 requires one pass over the data. In order to use the final sample to estimate statistics, we need to compute the Horvitz-Thompson inverse-probability estimator $\widehat{f(\nu_x)}$ for each of the sampled keys. Informally, the estimator for key $x$ in the sample is $f(\nu_x)/\Pr[x \text{ in sample}]$ (and 0 for keys not in the sample). To compute the estimator, we need to know the values $f(\nu_x)$ for the keys in the sample, which we get from a second pass over the data, and the conditional inclusion probabilities (the denominator), that have a closed form and can be computed. The parameter $\varepsilon$ trades off the running time of processing an element with the bound on the variance of the inverse-probability estimator.

We continue with an overview of the different components of the sketch. As mentioned above, we represent $f(\nu_x) = \mathcal{L}^c[a](\nu_x)_0^\gamma + \mathcal{L}^c[a](\nu_x)_\gamma^\infty$, and for each summand we maintain a separate sample of size $k$ (which will later be merged). For $\mathcal{L}^c[a](\nu_x)_0^\gamma$, we maintain a standard PPSWOR sketch. For $\mathcal{L}^c[a](\nu_x)_\gamma^\infty$, we build on a result from [8], which shows a way to map each input element into an temporary "output" element with a random value, such that if we look at all the output elements, $\mathsf{E}[\mathsf{Max}(x)] = \mathcal{L}^c[a](\nu_x)_\gamma^\infty$. These components were used in [8] to estimate the $f$-statistics of the data.

**Algorithm 4:** Produce a Final Sample from a Sampling Sketch Structure (Algorithm 3)

**Input:** sampling sketch structure $s$ for $f$
**Output:** sample of size $k$ of key and seed pairs
**if** $\int_\gamma^\infty a(t)dt > 0$ **then**
    **foreach** $e \in s.\mathsf{Sideline}$ **do**
        Process element
        $(e.key, \int_\gamma^\infty a(t)dt)$ by sketch
        $s.\mathsf{SumMax}$

**foreach** $e \in s.\mathit{SumMax}.\mathsf{sample}$ **do**
    $e.val \leftarrow r * e.val$ // `scale`
    `sample by` $r$
**if** $\int_0^\gamma ta(t)dt > 0$ **then**
    **foreach** $e \in s.ppswor.\mathsf{sample}$ **do**
        $e.val \leftarrow \frac{e.val}{\int_0^\gamma ta(t)dt}$
    $\mathsf{sample} \leftarrow$ merge
    $s.\mathsf{SumMax}.\mathsf{sample}$ and
    $s.ppswor.\mathsf{sample}$ // `bottom-`$k$
    `merge (Algorithm 1)`
**else**
    $\mathsf{sample} \leftarrow s.\mathsf{SumMax}.\mathsf{sample}$
**return** sample

However, in this work we need to produce a sample according to $\mathcal{L}^c[a](\nu_x)_\gamma^\infty$ (as opposed to estimating the sum of these quantities for all keys). In particular, when we look at the output elements, we only see their *random* value, but we are interested in producing a weighted sample according to their *expected* value. For that, we introduce the analysis of PPSWOR with stochastic inputs, which appears in Section 4. In that analysis, we establish the conditions that are needed in order for the sample according to the random values to be close to a sample according to the expected values.

The conditions in the analysis of stochastic PPSWOR require creating $k/\varepsilon$ independent output elements for each element we see, and subsequently, the sample we need for the range $\mathcal{L}^c[a](\nu_x)_\gamma^\infty$ is a PPSWOR sample of the output elements according to the weights $\mathsf{SumMax}(x)$ (defined in Section 5). That is the purpose of the SumMax sketch structure, which is presented in Section 5.

Each of the two samples we maintain (the PPSWOR and SumMax samples) has a fixed size and stores at most $k$ keys at any time. The $\gamma$ threshold is chosen to guarantee that we get the desired approximation ratio. The only structure that can use more space is the Sideline structure. As part of the analysis, we bound the size of the Sideline and show that in expectation, it is $O(k)$ and also provide worst case bounds on its maximum size during the run of the algorithm. The output elements that are processed by the SumMax sketch have a value that depends on $\gamma$ (which changes as we process the data), and the purpose of the Sideline structure is to store elements until $\gamma$ decreases enough that their value is fixed (and then they are removed from the Sideline and processed by the SumMax sketch).

The analysis results in the following main theorem.

**Theorem 3.1.** *Let $k \geq 3$ and $0 < \varepsilon \leq \frac{1}{2}$. Algorithms 3 and 4 produce a stochastic PPSWOR sample of size $k-1$, where each key $x$ has weight $V_x$ that satisfies $f(\nu_x) \leq \mathsf{E}[V_x] \leq \frac{1}{(1-\varepsilon)} f(\nu_x)$. The per-key inverse-probability estimator of $f(\nu_x)$ is unbiased and has variance*

$$\mathsf{Var}\left[\widehat{f(\nu_x)}\right] \leq \frac{4f(\nu_x)\sum_{z \in \mathcal{X}} f(\nu_z)}{(1-\varepsilon)^2(k-2)}.$$

*The space required by the sketch at any given time is $O(k)$ in expectation. Additionally, with probability at least $1 - \delta$, the space will not exceed $O\left(k + \min\{\log m, \log\log\left(\frac{\text{Sum}(W)}{\text{Min}(D)}\right)\} + \log\left(\frac{1}{\delta}\right)\right)$ at any time while processing $D$, where $m$ is the number of elements in $D$, $\text{Min}(D)$ is the minimum value of an element in $D$, and $\text{Sum}(W)$ is the sum of frequencies of all keys.*

## 4 Stochastic PPSWOR Sampling

In the PPSWOR sampling scheme described in Section 2.2, the weights $w_x$ of the keys were part of the deterministic input to the algorithm. In this section, we consider PPSWOR sampling when the weights are random variables. We will show that under certain assumptions, PPSWOR sampling according to randomized inputs is close to sampling according to the expected values of these random inputs.

Formally, let $\mathcal{X}$ be a set of keys. Each key $x \in \mathcal{X}$ is associated with $r_x \geq 0$ independent random variables $S_{x,1}, \ldots, S_{x,r_x}$ in the range $[0, T]$ (for some constant $T > 0$). The weight of key $x$ is the random variable $S_x := \sum_{i=1}^{r_x} S_{x,i}$. We additionally denote its expected weight by $v_x := \mathsf{E}[S_x]$, and the expected sum statistics by $V := \sum_x v_x$.

A *stochastic PPSWOR* sample is a PPSWOR sample computed for the key-value pairs $(x, S_x)$. That is, we draw the random variables $S_x$, then we draw for each $x$ a random variable $\texttt{seed}(x) \sim \mathsf{Exp}[S_x]$, and take the $k$ keys with lowest seed values.

The following result bounds the variance of estimating $v_x$ using a stochastic PPSWOR sample. We consider the conditional inverse-probability estimator of $v_x$. Note that even though the PPSWOR sample was computed using the random weight $S_x$, the estimator $\widehat{v_x}$ is computed using $v_x$ and will be $\frac{v_x}{\Pr[\texttt{seed}(x)<\tau]}$ for keys $x$ in the sample. It suffices to bound the per-key variance and relate it to the per-key variance bound for a PPSWOR sample computed directly for $v_x$. We show that when $V \geq Tk$, the overhead due to the stochastic sample is at most 4 (that is, the variance grows by a multiplicative factor of 4). The proof details would also reveal that when $V \gg Tk$, the worst-case bound on the overhead is actually closer to 2.

**Theorem 4.1.** *Let $k \geq 3$. In a stochastic PPSWOR sample, if $V \geq Tk$, then for every key $x \in \mathcal{X}$, the variance $\mathsf{Var}[\hat{v}_x]$ of the bottom-$k$ inverse probability estimator of $v_x$ is bounded by*

$$\mathsf{Var}[\hat{v}_x] \leq \frac{4v_x V}{k - 2}.$$

## 5 SumMax Sampling Sketch

We present an auxiliary sketch that processes elements $e = (e.key, e.val)$ with keys $e.key = (e.key.p, e.key.s)$ that are structured to have a primary key $e.key.p$ and a secondary key $e.key.s$. For each primary key $x$, we define

$$\mathsf{SumMax}_D(x) := \sum_{z|z.p=x} \mathsf{Max}_D(z)$$

where $\mathsf{Max}$ is as defined in Section 2. If there are no elements $e \in D$ such that $e.key.p = x$, then by definition $\mathsf{Max}_D(z) = 0$ for all $z$ with $z.p = x$ (as there are no elements in $D$ with key $z$) and therefore $\mathsf{SumMax}_D(x) = 0$. The $\mathsf{SumMax}$ sampling sketch (Algorithm 5) produces a PPSWOR sample of primary keys $x$ according to weights $\mathsf{SumMax}_D(x)$. Note that while the key space of the input elements contains structured keys of the form $e.key = (e.key.p, e.key.s)$, the key space for the output sample will be the space of primary keys only. The sketch structure consists of a bottom-$k$ structure and a hash function $h$. We assume we have a perfectly random hash function $h$ such that for every key $z = (z.p, z.s)$, $h(z) \sim \mathsf{Exp}[1]$ independently (in practice, we assume that the hash function is provided by the platform on which we run). We process an input element $e$ by generating a new data element with key $e.key.p$ (the primary key of the key of the input element) and value

$$\texttt{ElementScore}(e) := h(e.key)/e.val$$

and then processing that element by our bottom-$k$ structure. The bottom-$k$ structure holds our current sample of primary keys. By definition, the bottom-$k$ structure retains the $k$ primary keys $x$ with

**Algorithm 5:** SumMax sampling sketch

---

// **Initialize empty structure** $s$
**Input:** Sample size $k$
$s.h \leftarrow$ independent random hash with range $\mathsf{Exp}[1]$
Initialize $s.\mathsf{sample}$ // A bottom-$k$ structure (Algorithm 1)
// **Process element** $e = (e.key, e.val)$ where $e.key = (e.key.p, e.key.s)$
Process element $(e.key.p, s.h(e.key))/e.val)$ to structure $s.\mathsf{sample}$ // bottom-$k$ process element
// **Merge structures** $s_1, s_2$ (with $s_1.h = s_2.h$) to get $s$
$s.h \leftarrow s_1.h$ // $s_1.h = s_2.h$
$s.\mathsf{sample} \leftarrow$ Merge $s_1.\mathsf{sample}, s_2.\mathsf{sample}$// bottom-$k$ merge (Algorithm 1)

---

minimum

$$\mathtt{seed}_D(x) := \min_{e \in D | e.key.p = x} \mathtt{ElementScore}(e) \,.$$

To establish that this is a PPSWOR sample according to $\mathsf{SumMax}_D(x)$, we study the distribution of $\mathtt{seed}_D(x)$.

**Lemma 5.1.** *For all primary keys $x$ that appear in elements of $D$, $\mathtt{seed}_D(x) \sim Exp[\mathsf{SumMax}_D(x))]$. The random variables $\mathtt{seed}_D(x)$ are independent.*

Note that the distribution of $\mathtt{seed}_D(x)$, which is $\mathsf{Exp}[\mathsf{SumMax}_D(x)]$, does not depend on the particular structure of $D$ or the order in which elements are processed, but only on the parameter $\mathsf{SumMax}_D(x)$. The bottom-$k$ sketch structure maintains the $k$ primary keys with smallest $\mathtt{seed}_D(x)$ values. We therefore get the following corollary.

**Corollary 5.2.** *Given a stream or distributed set of elements $D$, the sampling sketch Algorithm 5 produces a PPSWOR sample according to the weights $\mathsf{SumMax}_D(x)$.*

## 6 Experiments

We implemented our sampling sketch in Python and report here the results of experiments on real and synthetic datasets. The implementation follows the pseudocode except that we incorporated two practical optimizations: removing redundant keys from the PPSWOR subsketch and removing redundant elements from Sideline. These optimizations do not affect the outcome of the computation or the worst-case analysis, but reduce the sketch size in practice. We used the following datasets:

- `abcnews` [23]: News headlines. For each word, we created an element with value $1$.
- `flicker` [31]: Tags used by Flickr users to annotate images. The key of each element is a tag, and the value is the number of times it appeared in a certain folder.
- Three synthetic generated datasets that contain $2 \times 10^6$ data elements. Each element has value $1$, and the key was chosen according to the Zipf distribution (numpy.random.zipf), with Zipf parameter values $\alpha \in \{1.1, 1.2, 1.5\}$. The Zipf family in this range is often a good model to real-world frequency distributions.

We applied our sampling sketch with sample size parameter values $k \in \{25, 50, 75, 100\}$ and set the parameter $\varepsilon = 0.5$ in all experiments. We sampled according to two concave sublinear functions: the frequency moment $f(\nu) = \nu^{0.5}$ and $f(\nu) = \ln(1 + \nu)$. Tables 1 reports aggregated results of 200 repetitions where we used the final sample to estimate the sum $\sum_{x \in \mathcal{X}} f(\nu_x)$. For error bounds, we list the worst-case bound on the CV (which depends only on $k$ and $\varepsilon$ and is $\propto 1/\sqrt{k}$) and report the actual normalized root of the average squared error (NRMSE). In addition, we report the NRMSE that we got from 200 repetitions of estimating the same statistics using two common sampling schemes for aggregated data, PPSWOR and priority sampling, which we use as benchmarks. We also consider the size of the sketch after processing each element. Since the representation of each key can be explicit and require a lot of space, we separately consider the number of distinct keys and the number of elements stored in the sketch. We report the maximum number of distinct keys stored in the sketch at any point (the average and the maximum over the 200 repetitions) and the respective maximum number of elements stored in the sketch at any point during the computations (again, the average and the maximum over the 200 repetitions). We can see that the actual error reported is significantly

Table 1: Experimental Results: $f(\nu) = \nu^{0.5}$ , $\ln(1+\nu)$

| $k$ | NRMSE | | Benchmark | | max #keys | | max #elem | |
|---|---|---|---|---|---|---|---|---|
| | bound | actual | ppswor | Pri. | ave | max | ave | max |
| | | | | $f(\nu) = \nu^{0.5}$, 200 reps | | | | |
| | | | Dataset: `abcnews` ($7.07 \times 10^6$ elements, $91.7 \times 10^3$ keys) | | | | | |
| 25 | 0.834 | 0.213 | 0.213 | 0.217 | 31.7 | 37 | 50.9 | 76 |
| 50 | 0.577 | 0.142 | 0.128 | 0.137 | 58.5 | 66 | 95.1 | 136 |
| 75 | 0.468 | 0.120 | 0.111 | 0.110 | 85.4 | 94 | 134.8 | 181 |
| 100 | 0.404 | 0.105 | 0.098 | 0.103 | 111.2 | 120 | 171.1 | 256 |
| | | | Dataset: `flickr` ($7.64 \times 10^6$ elements, $572.4 \times 10^3$ keys) | | | | | |
| 25 | 0.834 | 0.200 | 0.190 | 0.208 | 31.2 | 37 | 53.1 | 77 |
| 50 | 0.577 | 0.144 | 0.147 | 0.142 | 57.8 | 64 | 94.6 | 130 |
| 75 | 0.468 | 0.123 | 0.114 | 0.110 | 83.7 | 91 | 131.7 | 175 |
| 100 | 0.404 | 0.115 | 0.095 | 0.099 | 108.9 | 116 | 173.4 | 223 |
| | | | Dataset: `zipf1.1` ($2.00 \times 10^6$ elements, $652.2 \times 10^3$ keys) | | | | | |
| 25 | 0.834 | 0.215 | 0.198 | 0.217 | 31.8 | 39 | 52.5 | 75 |
| 50 | 0.577 | 0.123 | 0.137 | 0.131 | 58.7 | 66 | 95.0 | 130 |
| 75 | 0.468 | 0.109 | 0.115 | 0.114 | 84.7 | 91 | 135.2 | 186 |
| 100 | 0.404 | 0.106 | 0.103 | 0.097 | 111.2 | 119 | 176.3 | 221 |
| | | | Dataset: `zipf1.2` ($2.00 \times 10^6$ elements, $237.3 \times 10^3$ keys) | | | | | |
| 25 | 0.834 | 0.199 | 0.208 | 0.214 | 31.1 | 38 | 53.2 | 83 |
| 50 | 0.577 | 0.144 | 0.138 | 0.145 | 57.9 | 65 | 98.4 | 139 |
| 75 | 0.468 | 0.122 | 0.116 | 0.124 | 83.9 | 90 | 138.2 | 173 |
| 100 | 0.404 | 0.098 | 0.109 | 0.096 | 109.6 | 115 | 179.2 | 227 |
| | | | Dataset: `zipf1.5` ($2.00 \times 10^6$ elements, $22.3 \times 10^3$ keys) | | | | | |
| 25 | 0.834 | 0.201 | 0.207 | 0.194 | 30.1 | 35 | 53.4 | 74 |
| 50 | 0.577 | 0.152 | 0.139 | 0.142 | 56.1 | 60 | 101.5 | 136 |
| 75 | 0.468 | 0.115 | 0.115 | 0.112 | 81.6 | 86 | 151.8 | 199 |
| 100 | 0.404 | 0.098 | 0.094 | 0.086 | 107.1 | 113 | 196.3 | 248 |
| | | | $f(\nu) = \ln(1+\nu)$, 200 reps | | | | | |
| | | | Dataset: `abcnews` ($7.07 \times 10^6$ elements, $91.7 \times 10^3$ keys) | | | | | |
| 25 | 0.834 | 0.208 | 0.217 | 0.194 | 29.5 | 34 | 49.1 | 71 |
| 50 | 0.577 | 0.138 | 0.136 | 0.142 | 54.9 | 60 | 80.9 | 110 |
| 75 | 0.468 | 0.130 | 0.099 | 0.117 | 80.0 | 85 | 111.1 | 152 |
| 100 | 0.404 | 0.102 | 0.115 | 0.103 | 104.9 | 109 | 140.7 | 184 |
| | | | Dataset: `flickr` ($7.64 \times 10^6$ elements, $572.4 \times 10^3$ keys) | | | | | |
| 25 | 0.834 | 0.227 | 0.199 | 0.180 | 28.0 | 31 | 41.4 | 69 |
| 50 | 0.577 | 0.144 | 0.151 | 0.129 | 53.3 | 59 | 72.2 | 101 |
| 75 | 0.468 | 0.119 | 0.121 | 0.109 | 78.2 | 83 | 99.8 | 135 |
| 100 | 0.404 | 0.097 | 0.104 | 0.095 | 102.7 | 106 | 130.3 | 166 |
| | | | Dataset: `zipf1.1` ($2.00 \times 10^6$ elements, $652.2 \times 10^3$ keys) | | | | | |
| 25 | 0.834 | 0.201 | 0.204 | 0.234 | 29.2 | 34 | 48.8 | 71 |
| 50 | 0.577 | 0.127 | 0.132 | 0.129 | 54.4 | 58 | 80.4 | 119 |
| 75 | 0.468 | 0.116 | 0.122 | 0.110 | 79.6 | 84 | 110.9 | 142 |
| 100 | 0.404 | 0.107 | 0.106 | 0.104 | 104.5 | 109 | 139.8 | 165 |
| | | | Dataset: `zipf1.2` ($2.00 \times 10^6$ elements, $237.3 \times 10^3$ keys) | | | | | |
| 25 | 0.834 | 0.209 | 0.195 | 0.218 | 28.5 | 33 | 48.0 | 72 |
| 50 | 0.577 | 0.147 | 0.144 | 0.139 | 53.7 | 57 | 80.5 | 113 |
| 75 | 0.468 | 0.120 | 0.111 | 0.113 | 78.8 | 84 | 111.4 | 143 |
| 100 | 0.404 | 0.098 | 0.106 | 0.102 | 103.9 | 108 | 140.3 | 173 |
| | | | Dataset: `zipf1.5` ($2.00 \times 10^6$ elements, $22.3 \times 10^3$ keys) | | | | | |
| 25 | 0.834 | 0.210 | 0.197 | 0.226 | 27.2 | 30 | 45.2 | 66 |
| 50 | 0.577 | 0.141 | 0.146 | 0.149 | 52.1 | 55 | 78.9 | 104 |
| 75 | 0.468 | 0.124 | 0.112 | 0.106 | 76.9 | 79 | 110.5 | 146 |
| 100 | 0.404 | 0.100 | 0.101 | 0.099 | 101.9 | 104 | 139.1 | 173 |

lower than the worst-case bound. Furthermore, the error that our sketch gets is close to the error achieved by the two benchmark sampling schemes. We can also see that the maximum number of distinct keys stored in the sketch at any time is relatively close to the specified sample size of $k$ and that the total sketch size in terms of elements rarely exceeded $3k$, with the relative excess seeming to decrease with $k$. In comparison, the benchmark schemes require space that is the number of distinct keys (for the aggregation), which is significantly higher than the space required by our sketch.

# 7   Conclusion

We presented composable sampling sketches for weighted sampling of unaggregated data tailored to a concave sublinear function of the frequencies of keys. We experimentally demonstrated the simplicity and efficacy of our design: Our sketch size is nearly optimal in that it is not much larger than the final sample size, and the estimate quality is close to that provided by a weighted sample computed directly over the aggregated data.

## Acknowledgments

Ofir Geri was supported by NSF grant CCF-1617577, a Simons Investigator Award for Moses Charikar, and the Google Graduate Fellowship in Computer Science in the School of Engineering at Stanford University. The computing for this project was performed on the Sherlock cluster. We would like to thank Stanford University and the Stanford Research Computing Center for providing computational resources and support that contributed to these research results.

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
