[Supplementary Material]

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

The space required by our sketch significantly improves upon the previous methods (which all require aggregating the data). In particular, if the desired sample size is $k$, we show that the space required by the sketch at any given time is $O(k)$ in expectation. We additionally show that, with probability at least $1 - \delta$, the space will not exceed $O\left(k + \min\{\log m, \log \log \left(\frac{\mathsf{Sum}_D}{\mathsf{Min}(D)}\right)\} + \log \left(\frac{1}{\delta}\right)\right)$ at any time while processing the dataset $D$, where $m$ is the number of elements, $\mathsf{Sum}_D$ the sum of weights of all elements, and $\mathsf{Min}(D)$ is the minimum value of an element in $D$. In the common case where all elements have weight $1$, this means that for any $\delta$, the needed space is at most $O\left(k + \log \log m + \log \left(\frac{1}{\delta}\right)\right)$ with probability at least $1 - \delta$.[1]

We complement our work with a small-scale experimental study. We use a simple implementation of our sampling sketch to study the actual performance in terms of estimate quality and sketch size. In particular, we show that the estimate quality is even better than the (already adequate) guarantees provided by our worst-case bounds. We additionally compare the estimate quality to that of two popular sampling schemes for aggregated data, PPSWOR [37, 38] and priority (sequential Poisson) sampling [33, 17]. In the experiments, we see that the estimate quality of our sketch is close to what achieved by PPSWOR and priority sampling, while our sketch uses much less space by eliminating the need for aggregation.

The paper is organized as follows. The preliminaries are presented in Section 2. We provide an overview of PPSWOR and the statistical guarantees it provides for estimation. Then, we formally define the family of concave sublinear functions. Our sketch uses two building blocks. The first building block, which can be of independent interest, is the analysis of a *stochastic PPSWOR sample*. Typically, when computing a sample, the data from which we sample is a deterministic part of the input. In our construction, we needed to analyze the variance bounds for PPSWOR sampling that is computed over data elements with randomized weights (under certain assumptions). We provide this analysis in Section 3. The second building block is the *SumMax sampling sketch*, which is discussed in Section 4. This is an auxiliary sketch structure that supports datasets with a certain type of structured keys. We put it all together and describe our main result in Section 5. The experiments are discussed in Section 6.

**Related Work.**    There are multiple classic composable weighted sampling schemes for *aggregated* datasets (where keys are unique to elements). Schemes that provide estimators with tight worst-case variance bounds include priority (sequential Poisson) sampling [33, 17] and VarOpt sampling [8, 13]. We focus here on PPSWOR [37, 38] as our base scheme because it extends to *unaggregated* datasets, where multiple elements can additively contribute to the frequency/weight of each key.

There is a highly prolific line of research on developing sketch structures for different task over streamed or distributed unaggregated data with applications in multiple domains. Some early examples are frequent elements [31] and distinct counting [20], and the seminal work of [1] providing a theoretical model for frequency moments. Composable sampling sketches for unaggregated datasets were also studied for decades. The goal is to meet the quality of samples computed on aggregated

frequencies while using a sketch structure that can only hold a final-sample-size number of distinct keys. These include a folklore sketch for distinct sampling ($f(\nu) = 1$ when $\nu > 0$) [27, 39] and sketch structures for sum sampling ($f(\nu) = \nu$) [12]. The latter generalizes the discrete sample and hold scheme [22, 18, 14] and PPSWOR. Sampling sketches for cap functions ($f(\nu) = \min\{T, \nu\}$) were provided in [11] and have a slight overhead over the aggregated baseline. The latter work also provided multi-objective/universal samples that with a logarithmic overhead simultaneously provide statistical guarantees for all concave sublinear $f$. In the current work we propose sampling sketches that can be tailored to any concave sublinear function and only have a small constant overhead.

An important line of work uses sketches based on random linear projections to estimate frequency statistics and to sample. In particular, $\ell_p$ sampling sketches [21, 32, 2, 26, 25] sample (roughly) according to $f(\nu) = \nu^p$ for $p \in [0, 2]$. These sketches have a higher logarithmic overhead on the space compared to sample-based sketches, and do not support all concave sublinear functions of the frequencies (for example, $f(\nu) = \ln{(1 + \nu)}$). In some respects they are more limited in their application – for example, they are not designed to produce a sample that includes raw keys. Their advantage is that they can be used with super-linear ($p \in (1, 2]$) functions of frequencies and can also support signed element values (the turnstile model). For the more basic problem of sketches that estimate frequency statistics over the full data, a complete characterization of the frequency functions for which the statistics can be estimated via polylogarithmic-size sketches is provided in [6, 4]. Universal sketches for estimating $\ell_p$ norms of subsets were recently considered in [5]. The seminal work of Alon et al. [1] established that for some functions of frequencies (moments with $p > 2$), statistics estimation requires polynomial-size sketches. A double logarithmic size sketch, extending [19] for distinct counting, that computes statistics over the entire dataset for all soft concave sublinear functions is provided in [10]. Our design builds on components of that sketch.

## 2 Preliminaries

Consider a set $D$ of data elements of the form $e = (e.key, e.val)$ where $e.val > 0$. We denote the set of possible keys by $\mathcal{X}$. For a key $z \in \mathcal{X}$, we let $\mathsf{Max}_D(z) := \max_{e \in D | e.key = z} e.val$ and $\mathsf{Sum}_D(z) := \sum_{e \in D | e.key = z} e.val$ denote the maximum value of a data element in $D$ with key $z$ and the sum of values of data elements in $D$ with key $z$, respectively. Each key $z \in \mathcal{X}$ that appears in $D$ is called *active*. If there is no element $e \in D$ with $e.key = z$, we say that $z$ is *inactive* and define $\mathsf{Max}_D(z) := 0$ and $\mathsf{Sum}_D(z) := 0$. When $D$ is clear from context, it is omitted. For a key $z$, we use the shorthand $\nu_z := \mathsf{Sum}_D(z)$ and refer to it as the frequency of $z$. The sum and the *max-distinct statistics* of $D$ are defined, respectively, as $\mathsf{Sum}_D := \sum_{e \in D} e.val$ and $\mathsf{MxDistinct}_D := \sum_{z \in \mathcal{X}} \mathsf{Max}_D(z)$. For a function $f$, $f_D := \sum_{z \in \mathcal{X}} f(\mathsf{Sum}_D(z)) = \sum_{z \in \mathcal{X}} f(\nu_z)$ is the *f-frequency statistics* of $D$.

For a set $A \subseteq \mathbb{R}$, we use $A_{(i)}$ to denote the $i$-th order statistic of $A$, that is, the $i$-th lowest element in $A$.

### 2.1 The Composable Bottom-$k$ Structure

In this work, we will use composable sketch structures in order to efficiently summarize streamed or distributed data elements. A composable sketch structure is specified by three operations: The initialization of an empty sketch structure $s$, the processing of a data element $e$ into a structure $s$, and the merging of two sketch structures $s_1$ and $s_2$. To sketch a stream of elements, we start with an empty structure and sequentially process data elements while storing only the sketch structure. The merge operation is useful with distributed or parallel computation and allows us to compute the sketch of a large set $D = \bigcup_i D_i$ of data elements by merging the sketches of the parts $D_i$.

In particular, one of the main building blocks that we use is the bottom-$k$ structure [15], specified in Algorithm 1. The structure maintains $k$ data elements: For each key, consider only the element with that key that has the minimum value. Of these elements, the structure keeps the $k$ elements that have the lowest values.

### 2.2 The PPSWOR Sampling Sketch

In this subsection, we describe a scheme to produce a sample of $k$ keys, where at each step the probability that a key is selected is proportional to its weight. That is, the sample we produce will

**Algorithm 1:** Bottom-$k$ Sketch Structure

---

**// Initialize structure**
**Input:** the structure size $k$
$s.set \leftarrow \emptyset$ // Set of $\leq k$ key-value pairs
**// Process element**
**Input:** element $e = (e.key, e.val)$, a bottom-$k$ structure $s$
**if** $e.key \in s.set$ **then**
| replace the current value $v$ of $e.key$ in $s.set$ with $\min\{v, e.val\}$
**else**
| insert $(e.key, e.val)$ to $s.set$
| **if** $|s.set| = k + 1$ **then**
| | Remove the element $e'$ with maximum value from $s.set$

**// Merge two bottom-$k$ structures**
**Input:** $s_1, s_2$ // Bottom-$k$ structures
**Output:** $s$ // Bottom-$k$ structure
$P \leftarrow s_1.set \cup s_2.set$
$s.set \leftarrow$ the (at most) $k$ elements of $P$ with lowest values (at most one element per key)

---

**Algorithm 2:** PPSWOR Sampling Sketch

---

**// Initialize structure**
**Input:** the sample size $k$
Initialize a bottom-$k$ structure $s$.sample // Algorithm 1
**// Process element**
**Input:** element $e = (e.key, e.val)$, PPSWOR sample structure $s$
$v \sim \mathsf{Exp}[e.val]$
Process the element $(e.key, v)$ into the bottom-$k$ structure $s$.sample
**// Merge two structures** $s_1, s_2$ to obtain $s$
$s$.sample $\leftarrow$ Merge the bottom-$k$ structures $s_1$.sample and $s_2$.sample

---

be equivalent to performing the following $k$ steps. At each step we select one key and add it to the sample. At the first step, each key $x \in \mathcal{X}$ (with weight $w_x$) is selected with probability $w_x / \sum_y w_y$. At each subsequent step, we choose one of the remaining keys, again with probability proportional to its weight. Since the total weight of the remaining keys is lower, the probability that a key is selected in a subsequent step (provided it was not selected earlier) is only higher. This process is called probability proportional to size and without replacement (PPSWOR) sampling.

A classic method for PPSWOR sampling is the following scheme [37, 38]. For each key $x$ with weight $w_x$, we independently draw $\mathtt{seed}(x) \sim \mathsf{Exp}(w_x)$. Outputting the sample that includes the $k$ keys with smallest $\mathtt{seed}(x)$ is equivalent to PPSWOR sampling as described above. This method together with a bottom-$k$ structure can be used to implement PPSWOR sampling over a set of data elements $D$ according to $\nu_x = \mathsf{Sum}_D(x)$. This sampling sketch is presented here as Algorithm 2. The sketch is due to [12] (based on [22, 18, 14]).

**Proposition 2.1.** *Algorithm 2 maintains a composable bottom-$k$ structure such that for each key $x$, the lowest value of an element with key $x$ (denoted by $\mathtt{seed}(x)$) is drawn independently from $\mathsf{Exp}(\nu_x)$. Hence, it is a PPSWOR sample according to the weights $\nu_x$.*

The proof is provided in Appendix A.

## 2.3 Estimation Using Bottom-$k$ Samples

PPSWOR sampling (Algorithm 2) is a special case of bottom-$k$ sampling [38, 15, 16].

**Definition 2.2.** *Let $k \geq 2$. A* bottom-$k$ sample *over keys $\mathcal{X}$ is obtained by drawing independently for each active key $x$ a random variable $\mathtt{seed}(x) \sim \mathsf{SeedDist}_x$. The $k - 1$ keys with lowest $\mathtt{seed}(x)$ values are considered to be included in the sample $S$, and the $k$-th lowest value $\tau := \{\mathtt{seed}(x) \mid x \in \mathcal{X}\}_{(k)}$ is the* inclusion threshold.

The distributions $\mathsf{SeedDist}_x$ are such that for all $t > 0$, $\Pr[\mathsf{seed}(x) < t] > 0$, that is, there is positive probability to be below any positive $t$. Typically the distributions $\mathsf{SeedDist}_x$ come from a family of distributions that is parameterized by the *frequency* $\nu_x$ of keys. The frequency is positive for active keys and $0$ otherwise. In the special case of PPSWOR sampling by frequency, $\mathsf{SeedDist}_x$ is $\mathsf{Exp}(\nu_x)$.

We review here how a bottom-$k$ sample is used to estimate domain statistics of the form $\sum_{x \in H} f(\nu_x)$ for $H \subseteq \mathcal{X}$. More generally, we will show how to estimate aggregates of the form

$$\sum_{x \in \mathcal{X}} L_x f(\nu_x) \tag{1}$$

for any set of fixed values $L_x$. Note that we can represent domain statistics in this form by setting $L_x = 1$ for $x \in H$ and $L_x = 0$ for $x \notin H$. For the sake of this discussion, we treat $f_x := f(\nu_x)$ as simply a set of weights associated with keys, assuming we can have $f_x > 0$ only for active keys.

In order to estimate statistics of the form (1), we will define an estimator $\widehat{f}_x$ for each $f_x$. The estimator $\widehat{f}_x$ will be non-negative ($\widehat{f}_x \geq 0$), unbiased ($\mathsf{E}\left[\widehat{f}_x\right] = f_x$), and such that $\widehat{f}_x = 0$ when the key $x$ is not included in the bottom-$k$ sample ($x \notin S$ using the terms of Definition 2.2).

As a general convention, we will use the notation $\widehat{z}$ to denote an estimator of any quantity $z$. We define the *sum estimator* of the statistics $\sum_{x \in \mathcal{X}} L_x f_x$ to be

$$\widehat{\sum_{x \in \mathcal{X}} L_x f_x} := \sum_{x \in S} L_x \widehat{f}_x$$

We also note that since $\widehat{f}_x = 0$ for $x \notin S$, computing the sum only over $x \in S$ is the same as computing the sum over all $x \in \mathcal{X}$, that is, $\sum_{x \in S} L_x \widehat{f}_x = \sum_{x \in \mathcal{X}} L_x \widehat{f}_x$.

Note that the sum estimator can be computed as long as the fixed values $L_x$ and the per-key estimates $\widehat{f}_x$ for $x \in S$ are available. From linearity of expectation, we get that the sum estimate is unbiased:

$$\mathsf{E}\left[\widehat{\sum_{x \in \mathcal{X}} L_x f_x}\right] = \mathsf{E}\left[\sum_{x \in S} L_x \widehat{f}_x\right]$$

$$= \mathsf{E}\left[\sum_{x \in \mathcal{X}} L_x \widehat{f}_x\right] = \sum_{x \in \mathcal{X}} L_x \mathsf{E}\left[\widehat{f}_x\right] = \sum_{x \in \mathcal{X}} L_x f_x.$$

We now define the per-key estimators $\widehat{f}_x$. The following is a conditioned variant of the Horvitz-Thompson estimator [24].

**Definition 2.3.** *Let $k \geq 2$ and consider a bottom-$k$ sample, where $S$ is the set of $k-1$ keys in the sample and $\tau$ is the inclusion threshold. For any $x \in \mathcal{X}$, the* inverse-probability estimator *of $f_x$ is*

$$\widehat{f}_x = \begin{cases} \frac{f_x}{\Pr_{\mathsf{seed}(x) \sim \mathsf{SeedDist}_x}[\mathsf{seed}(x) < \tau]} & x \in S \\ 0 & x \notin S \end{cases}.$$

In order to compute these estimates, we need to know the weights $f_x$ and distributions $\mathsf{SeedDist}_x$ for the sampled keys $x \in S$. In particular, in our applications when $f_x = f(\nu_x)$ is a function of frequency and the seed distribution is parameterized by frequency, then it suffices to know the frequencies of sampled keys.[2]

**Claim 2.4.** *The inverse-probability estimator is unbiased, that is, $\mathsf{E}\left[\widehat{f}_x\right] = f_x$.*

*Proof.* We first consider $\widehat{f}_x$ when conditioned on the seed values of all other keys $\mathcal{X} \setminus \{x\}$ and in particular on

$$\tau_x := \{\mathsf{seed}(z) \mid z \in \mathcal{X} \setminus \{x\}\}_{(k-1)},$$

which is the $k-1$ smallest seed on $\mathcal{X} \setminus \{x\}$. Under this conditioning, a key $x$ is included in $S$ with probability $\Pr_{\mathtt{seed}(x) \sim \mathsf{SeedDist}_x}[\mathtt{seed}(x) < \tau_x]$. When $x \notin S$, the estimate is 0. When $x \in S$, we have that $\tau_x = \tau$ and the estimate is the ratio of $f_x$ and the inclusion probability. So our estimator is a plain inverse probability estimator and thus $\mathsf{E}\left[\widehat{f_x} \mid \tau_x\right] = f_x$.

Finally, from the fact that the estimator is unbiased when conditioned on $\tau_x$, we also get that it is unconditionally unbiased: $\mathsf{E}\left[\widehat{f_x}\right] = \mathsf{E}_{\tau_x}\left[\mathsf{E}\left[\widehat{f_x} \mid \tau_x\right]\right] = \mathsf{E}_{\tau_x}\left[f_x\right] = f_x$. $\qquad\square$

We now turn to analyze the variance of the estimators. The guarantees we can obtain on the quality of the sum estimates depend on how well the distributions $\mathsf{SeedDist}_x$ are tailored to the values $f_x$, where ideally, keys should be sampled with probabilities proportional to $f_x$. PPSWOR, where $\mathtt{seed}(x) \sim \mathsf{Exp}(f_x)$, is such a "gold standard" sampling scheme that provides us with strong guarantees: For domain statistics $\sum_{x \in H} f_x$, we get a tight worst-case bound on the coefficient of variation[3] of $1/\sqrt{q(k-2)}$, where $q = \sum_{x \in H} f_x / \sum_{x \in \mathcal{X}} f_x$ is the fraction of the statistics that is due to the domain $H$. Moreover, the estimates are concentrated in a Chernoff bounds sense. For objectives of the form (1), we obtain additive Hoeffding-style bounds that depend only on sample size and the range of $L_x$.

When we cannot implement "gold-standard" sampling via small composable sampling structures, we seek guarantees that are close to that. Conveniently, in the analysis it suffices to bound the variance of the *per-key* estimators [9, 11]: A key property of bottom-$k$ estimators is that $\forall x, z$, $\mathsf{cov}(\widehat{f_x}, \widehat{f_z}) \leq 0$ (equality holds for $k \geq 3$) [11]. Therefore, the variance of the sum estimator can be bounded by the sum of bounds on the per-key variance. This allows us to only analyze the per-key variance $\mathsf{Var}\left(\widehat{f_x}\right)$.

To achieve the guarantees of the "gold standard" sampling, the desired bound on the per-key variance for a sample of size $k-1$ (a bottom-$k$ sample where the $k$-th lowest seed is the inclusion threshold) is

$$\mathsf{Var}\left(\widehat{f_x}\right) \leq \frac{1}{k-2} f_x \sum_{z \in \mathcal{X}} f_z . \tag{2}$$

So our goal is to establish upper bounds on the per-key variance that are within a small constant of (2). We refer to this value as the *overhead*. The overhead factor in the per-key bounds carries over to the sum estimates.

We next review the methodology for deriving per-key variance bounds. The starting point is to first bound the per-key variance of $\widetilde{f_x}$ conditioned on $\tau_x$.

**Claim 2.5.** *With the inverse probability estimator we have*

$$\mathsf{Var}\left(\widehat{f_x}\right) = \mathsf{E}_{\tau_x}\left[\mathsf{Var}\left(\widehat{f_x} \mid \tau_x\right)\right] = \mathsf{E}_{\tau_x}\left[f_x^2\left(\frac{1}{\Pr[\mathtt{seed}(x) < \tau_x]} - 1\right)\right] .$$

*Proof.* Follows from the law of total variance and the unbiasedness of the conditional estimates for any fixed value of $\tau_x$, $\mathsf{E}\left[\widehat{f_x} \mid \tau_x\right] = f_x$. $\qquad\square$

For the "gold standard" PPSWOR sample, we have $\Pr[\mathtt{seed}(x) < t] = 1 - \exp(-f_x t)$ and using $\frac{e^{-x}}{1-e^{-x}} \leq \frac{1}{x}$, we get

$$\mathsf{Var}\left(\widehat{f_x} \mid \tau_x\right) = f_x^2\left(\frac{1}{\Pr[\mathtt{seed}(x) < \tau_x]} - 1\right) \leq \frac{f_x}{\tau_x} . \tag{3}$$

In order to bound the unconditional per-key variance, we use the following notion of stochastic dominance.

**Definition 2.6.** *Consider two density functions $a$ and $b$ both with support on the nonnegative reals. We say that $a$ is* dominated by $b$ ($a \preceq b$) *if for all $z \geq 0$, $\int_0^z a(y)dy \leq \int_0^z b(y)dy$.*

Note that since they are both density functions, it implies that the CDF of $a$ is pointwise at most the CDF of $b$. In particular, the probability of being below some value $y$ under $b$ is at least that of $a$.[4]

When bounding the variance, we use a distribution $B$ that dominates the distribution of $\tau_x$ and is easier to work with and then compute the upper bound

$$\mathsf{Var}\left[\widehat{f}_x\right] = \mathsf{E}_{\tau_x}\left[f_x^2\left(\frac{1}{\Pr[\mathtt{seed}(x) < \tau_x]} - 1\right)\right] \tag{4}$$

$$\leq \mathsf{E}_{t\sim B}\left[f_x^2\left(\frac{1}{\Pr[\mathtt{seed}(x) < t]} - 1\right)\right] .$$

With PPSWOR, the distribution of $\tau_x$ is dominated by the distribution $\mathsf{Erlang}[\sum_{z\in\mathcal{X}} f_z, k-1]$, where $\mathsf{Erlang}[V,k]$ is the distribution of the sum of $k$ independent exponential random variables with parameter $V$. The density function of $\mathsf{Erlang}[V,k]$ is $B_{V,k}(t) = \frac{V^k t^{k-1} e^{-Vt}}{(k-1)!}$. Choosing $B$ to be $\mathsf{Erlang}[\sum_{z\in\mathcal{X}} f(\nu_z), k-1]$ in (4) and using (3), we get the bound in (2).

Note that if we have an estimator that gives a weaker bound of $c \cdot \frac{f_x}{\tau_x}$ on the conditional variance and the distribution of $\tau_x$ is similarly dominated by $\mathsf{Erlang}[\sum_{z\in\mathcal{X}} f_z, k-1]$, we will obtain a corresponding bound on the unconditional variance with overhead $c$.

## 2.4 Concave Sublinear Functions

A function $f : [0,\infty) \to [0,\infty)$ is *soft concave sublinear* if for some $a(t) \geq 0$ it can be expressed as[5]

$$f(\nu) = \mathcal{L}^{\mathrm{c}}[a](\nu) := \int_0^\infty a(t)(1 - e^{-\nu t})dt . \tag{5}$$

$\mathcal{L}^{\mathrm{c}}[a](\nu)$ is called the *complement Laplace transform* of $a$ at $\nu$. The function $a(t)$ is the inverse Laplace$^c$ (complement Laplace) transform of $f$:

$$a(t) = (\mathcal{L}^{\mathrm{c}})^{-1}[f](t) . \tag{6}$$

A table with the inverse Laplace$^c$ transform of several common functions (in particular, the moments $\nu^p$ for $p \in (0,1)$ and $\ln(1+\nu)$) appears in [10]. We additionally use the notation

$$\mathcal{L}^{\mathrm{c}}[a](\nu)_\alpha^\beta := \int_\alpha^\beta a(t)(1 - e^{-\nu t})dt .$$

The sampling schemes we present in this work will be defined for *soft* concave sublinear functions of the frequencies. However, this will allow us to estimate well any function that is within a small multiplicative constant of a soft concave sublinear function. In particular, we can estimate *concave sublinear functions*. These functions can be expressed as

$$f(\nu) = \int_0^\infty a(t)\min\{1, \nu t\}dt \tag{7}$$

for $a(t) \geq 0$.[6] The concave sublinear family includes all functions such that $f(0) = 0$, $f$ is monotonically non-decreasing, $\partial_+ f(0) < \infty$, and $\partial^2 f \leq 0$.

Any concave sublinear function $f$ can be approximated by a soft concave sublinear function as follows. Consider the corresponding soft concave sublinear function $\tilde{f}$ using the same coefficients $a(t)$. The function $\tilde{f}$ closely approximates $f$ pointwise [10]:

$$(1 - 1/e)f(\nu) \leq \tilde{f}(\nu) \leq f(\nu) .$$

Our weighted sample for $\tilde{f}$ will respectively approximate a weighted sample for $f$ (later explained in Remark 5.11).

## 3 Stochastic PPSWOR Sampling

In this section, we provide an analysis of PPSWOR for a case that will appear later in our main sketch. The case we consider is the following. In the PPSWOR sampling scheme described in Section 2.2, the weights $w_x$ of the keys were part of the deterministic input to the algorithm. In this section, we consider PPSWOR sampling when the weights are random variables. We will show that under certain assumptions, PPSWOR sampling according to randomized inputs is close to sampling according to the expected values of these random inputs.

Formally, let $\mathcal{X}$ be a set of keys. Each key $x \in \mathcal{X}$ is associated with $r_x \geq 0$ independent random variables $S_{x,1}, \ldots, S_{x,r_x}$ in the range $[0, T]$ (for some constant $T > 0$). The weight of key $x$ is the random variable $S_x := \sum_{i=1}^{r_x} S_{x,i}$. We additionally denote its expected weight by $v_x := \mathsf{E}[S_x]$, and the expected sum statistics by $V := \sum_x v_x$.

A *stochastic PPSWOR* sample is a PPSWOR sample computed for the key-value pairs $(x, S_x)$. That is, we draw the random variables $S_x$, then we draw for each $x$ a random variable $\mathtt{seed}(x) \sim \mathsf{Exp}[S_x]$, and take the $k$ keys with lowest seed values.

We prove two results that relate stochastic PPSWOR sampling to a PPSWOR sample according to the expected values $v_x$. The first result bounds the variance of estimating $v_x$ using a stochastic PPSWOR sample. We consider the conditional inverse-probability estimator of $v_x$ (Definition 2.3). Note that even though the PPSWOR sample was computed using the random weight $S_x$, the estimator $\widehat{v_x}$ is computed using $v_x$ and will be $\frac{v_x}{\Pr[\mathtt{seed}(x)<\tau]}$ for keys $x$ in the sample. Based on the discussion in Section 2.3, it suffices to bound the per-key variance and relate it to the per-key variance bound for a PPSWOR sample computed directly for $v_x$. We show that when $V \geq Tk$, the overhead due to the stochastic sample is at most $4$ (that is, the variance grows by a multiplicative factor of $4$). The proof details would also reveal that when $V \gg Tk$, the worst-case bound on the overhead is actually closer to $2$.

**Theorem 3.1.** *Let $k \geq 3$. In a stochastic PPSWOR sample, if $V \geq Tk$, then for every key $x \in \mathcal{X}$, the variance $\mathsf{Var}[\hat{v}_x]$ of the bottom-$k$ inverse probability estimator of $v_x$ is bounded by*

$$\mathsf{Var}[\hat{v}_x] \leq \frac{4v_x V}{k-2}.$$

Note that in order to compute these estimates, we need to be able to compute the values $v_x = \mathsf{E}[S_x]$ and $\Pr[\mathtt{seed}(x) < \tau]$ for sampled keys. With stochastic sampling, the precise distribution $\mathsf{SeedDist}_x$ depends on the distributions of the random variables $S_{x,i}$. For now, however, we assume that $\mathsf{SeedDist}_x$ and $v_x$ are available to us with the sample. In Section 5, when we use stochastic sampling, we will also show how to compute $\mathsf{SeedDist}_x$.

The second result in this section provides a lower bound on the probability that a key $x$ is included in the stochastic PPSWOR sample of size $k = 1$. We show that when $V \geq \frac{1}{\varepsilon} \ln\left(\frac{1}{\varepsilon}\right) T$, the probability key $x$ is included in the sample is at least $1 - 2\varepsilon$ times the probability it is included in a PPSWOR sample according to the expected weights.

**Theorem 3.2.** *Let $\varepsilon \leq \frac{1}{2}$. Consider a stochastic PPSWOR sample of size $k = 1$. If $V \geq \frac{1}{\varepsilon} \ln\left(\frac{1}{\varepsilon}\right) T$, the probability that any key $x \in \mathcal{X}$ is included in the sample is at least $(1 - 2\varepsilon)\frac{v}{V}$.*

The proofs of the two theorems are deferred to Appendix B.

## 4 SumMax Sampling Sketch

In this section, we present an auxiliary sampling sketch which will be used in Section 5. The sketch processes elements $e = (e.key, e.val)$ with keys $e.key = (e.key.p, e.key.s)$ that are structured to have a primary key $e.key.p$ and a secondary key $e.key.s$. For each primary key $x$, we define

$$\mathsf{SumMax}_D(x) := \sum_{z | z.p = x} \mathsf{Max}_D(z)$$

---

**Algorithm 3:** SumMax Sampling Sketch

---

// **Initialize empty structure** $s$
**Input:** Sample size $k$
$s.h \leftarrow$ fully independent random hash with range $\mathsf{Exp}[1]$
Initialize $s.\mathsf{sample}$ // A bottom-$k$ structure (Algorithm 1)
// **Process element** $e = (e.key, e.val)$ where $e.key = (e.key.p, e.key.s)$
Process element $(e.key.p, s.h(e.key)/e.val)$ to structure $s.\mathsf{sample}$ // bottom-$k$ process
  element (Algorithm 1)
// **Merge structures** $s_1$, $s_2$ (with $s_1.h = s_2.h$) to get $s$
$s.h \leftarrow s_1.h$ // $s_1.h = s_2.h$
$s.\mathsf{sample} \leftarrow \mathsf{Merge}\ s_1.\mathsf{sample}, s_2.\mathsf{sample}$// bottom-$k$ merge (Algorithm 1)

---

where $\mathsf{Max}$ is as defined in Section 2. If there are no elements $e \in D$ such that $e.key.p = x$, then by definition $\mathsf{Max}_D(z) = 0$ for all $z$ with $z.p = x$ (as there are no elements in $D$ with key $z$) and therefore $\mathsf{SumMax}_D(x) = 0$. Our goal in this section is to design a sketch that produces a PPSWOR sample of primary keys $x$ according to weights $\mathsf{SumMax}_D(x)$. Note that while the key space of the input elements contains structured keys of the form $e.key = (e.key.p, e.key.s)$, the key space for the output sample will be the space of primary keys only. Our sampling sketch is described in Algorithm 3.

The sketch structure consists of a bottom-$k$ structure and a hash function $h$. We assume we have a perfectly random hash function $h$ such that for every key $z = (z.p, z.s)$, $h(z) \sim \mathsf{Exp}[1]$ independently (in practice, we assume that the hash function is provided by the platform on which we run). We process an input element $e$ by generating a new data element with key $e.key.p$ (the primary key of the key of the input element) and value

$$\texttt{ElementScore}(e) := h(e.key)/e.val$$

and then processing that element by our bottom-$k$ structure. The bottom-$k$ structure holds our current sample of primary keys.

By definition, the bottom-$k$ structure retains the $k$ primary keys $x$ with minimum

$$\texttt{seed}_D(x) := \min_{e \in D | e.key.p = x} \texttt{ElementScore}(e) \,.$$

To establish that this is a PPSWOR sample according to $\mathsf{SumMax}_D(x)$, we study the distribution of $\texttt{seed}_D(x)$.

**Lemma 4.1.** *For all primary keys $x$ that appear in elements of $D$, $\texttt{seed}_D(x) \sim Exp[\mathsf{SumMax}_D(x)]$. The random variables $\texttt{seed}_D(x)$ are independent.*

The proof is deferred to Appendix C.

Note that the distribution of $\texttt{seed}_D(x)$, which is $\mathsf{Exp}[\mathsf{SumMax}_D(x)]$, does not depend on the particular structure of $D$ or the order in which elements are processed, but only on the parameter $\mathsf{SumMax}_D(x)$. The bottom-$k$ sketch structure maintains the $k$ primary keys with smallest $\texttt{seed}_D(x)$ values. We therefore get the following corollary.

**Corollary 4.2.** *Given a stream or distributed set of elements $D$, the sampling sketch in Algorithm 3 produces a PPSWOR sample according to the weights $\mathsf{SumMax}_D(x)$.*

## 5 Sampling Sketch for Functions of Frequencies

In this section, we are given a set $D$ of elements $e = (e.key, e.val)$ and we wish to maintain a sample of $k$ keys, that will be close to PPSWOR according to a soft concave sublinear function of their frequencies $f(\nu_x)$. At a high level, our sampling sketch is guided by the sketch for estimating the statistics $f_D$ due to Cohen [10]. Our sketch uses a parameter $\varepsilon$ that will tradeoff the running time of processing an element with the bound on the variance of the inverse-probability estimator.

Recall that a soft concave sublinear function $f$ can be represented as $f(w) = \mathcal{L}^c[a](w)_0^\infty = \int_0^\infty a(t)(1 - e^{-wt})dt$ for $a(t) \geq 0$. Using this representation, we express $f(\nu_x)$ as a sum of two

contributions for each key $x$:

$$f(\nu_x) = \mathcal{L}^c[a](\nu_x)_0^\gamma + \mathcal{L}^c[a](\nu_x)_\gamma^\infty,$$

where $\gamma$ is a value we will set adaptively while processing the elements. Our sampling sketch is described in Algorithm 4. It maintains a separate sampling sketch for each set of contributions. The sketch for $\mathcal{L}^c[a](\nu_x)_0^\gamma$ is discussed in Section 5.1, and the sketch for $\mathcal{L}^c[a](\nu_x)_\gamma^\infty$ is discussed in Section 5.2. In order to produce a sample from the sketch, these separate sketches need to be combined. Algorithm 5 describes how to produce a final sample from the sketch. This is discussed further in Section 5.3. Finally, we discuss the computation of the inverse-probability estimators $\widehat{f(\nu_x)}$ for the sampled keys in Section 5.4. In particular, in order to compute the estimator, we need to know the values $f(\nu_x)$ for the keys in the sample, which will require a second pass over the data. The analysis will result in the following main theorem.

---

**Algorithm 4:** Sampling Sketch Structure for $f$

---

**// Initialize empty structure** $s$
**Input:** $k$: Sample size, $\varepsilon$, $a(t) \geq 0$
Initialize $s.\mathsf{SumMax}$ // $\mathsf{SumMax}$ sketch of size $k$ (Algorithm 3)
Initialize $s.ppswor$ // PPSWOR sketch of size $k$ (Algorithm 2)
Initialize $s.sum \leftarrow 0$ // A sum of all the elements seen so far
Initialize $s.\gamma \leftarrow \infty$ // Threshold
Initialize $s.\mathsf{Sideline}$ // A composable max-heap/priority queue
**// Process element**
**Input:** Element $e = (e.key, e.val)$, structure $s$
Process $e$ by $s.ppswor$
$s.sum \leftarrow s.sum + e.val$
$s.\gamma \leftarrow \frac{2\varepsilon}{s.sum}$
// $r = k/\varepsilon$
**foreach** $i \in [r]$ **do**
    $y \sim \mathsf{Exp}[e.val]$ // Exponentially distributed with parameter $e.val$
    // Process in $\mathsf{Sideline}$
    **if** *The key* $(e.key, i)$ *appears in* $s.\mathsf{Sideline}$ **then**
        Update the value of $(e.key, i)$ to be the minimum of $y$ and the current value
    **else**
        Add the element $((e.key, i), y)$ to $s.\mathsf{Sideline}$

**while** $s.\mathsf{Sideline}$ *contains an element* $g = (g.key, g.val)$ *with* $g.val \geq s.\gamma$ **do**
    Remove $g$ from $s.\mathsf{Sideline}$
    **if** $\int_{g.val}^\infty a(t)dt > 0$ **then**
        Process element $(g.key, \int_{g.val}^\infty a(t)dt)$ by $s.\mathsf{SumMax}$

**// Merge two structures** $s_1$ and $s_2$ to $s$ (with same $k, \varepsilon, a$ and same $h$ in
    $\mathsf{SumMax}$ sub-structures)
$s.sum \leftarrow s_1.sum + s_2.sum$
$s.\gamma \leftarrow \frac{2\varepsilon}{s.sum}$
$s.\mathsf{Sideline} \leftarrow$ merge $s1.\mathsf{Sideline}$ and $s2.\mathsf{Sideline}$ // Merge priority queues.
$s.ppswor \leftarrow$ merge $s_1.ppswor$ and $s_2.ppswor$ // Merge PPSWOR structures
$s.\mathsf{SumMax} \leftarrow$ merge $s_1.\mathsf{SumMax}$ and $s_2.\mathsf{SumMax}$ // Merge SumMax structures
**while** $s.\mathsf{Sideline}$ *contains an element* $g = (g.key, g.val)$ *with* $g.val \geq s.\gamma$ **do**
    Remove $g$ from $s.\mathsf{Sideline}$
    **if** $\int_{g.val}^\infty a(t)dt > 0$ **then**
        Process element $(g.key, \int_{g.val}^\infty a(t)dt)$ by $s.\mathsf{SumMax}$

---

**Theorem 5.1.** *Let $k \geq 3$, $0 < \varepsilon \leq \frac{1}{2}$, and $f$ be a soft concave sublinear function. Algorithms 4 and 5 produce a stochastic PPSWOR sample of size $k - 1$, where each key $x$ has weight $V_x$ that satisfies $f(\nu_x) \leq \mathsf{E}[V_x] \leq \frac{1}{(1-\varepsilon)}f(\nu_x)$. The per-key inverse-probability estimator of $f(\nu_x)$ is unbiased and*

**Algorithm 5:** Produce a Final Sample from a Sampling Sketch Structure (Algorithm 4)

---

**Input:** Sampling sketch structure $s$ for $f$
**Output:** Sample of size $k$ of key and seed pairs
**if** $\int_\gamma^\infty a(t)dt > 0$ **then**
    **foreach** $e \in s.$Sideline **do**
        Process element $(e.key, \int_\gamma^\infty a(t)dt)$ by sketch $s.$ SumMax

**foreach** $e \in s.$ *SumMax*.sample **do**
    $e.val \leftarrow r * e.val$ // Multiply value by $r$

**if** $\int_0^\gamma ta(t)dt > 0$ **then**
    **foreach** $e \in s.ppswor.$sample **do**
        $e.val \leftarrow \frac{e.val}{\int_0^\gamma ta(t)dt}$ // Divide value by $B(\gamma)$
    sample $\leftarrow$ merge $s.$ SumMax .sample and $s.ppswor.$sample // Bottom-$k$ merge
    (Algorithm 1)
**else**
    sample $\leftarrow s.$ SumMax .sample
**return** sample

---

*has variance*

$$\mathsf{Var}\left[\widehat{f(\nu_x)}\right] \leq \frac{4f(\nu_x)\sum_{z\in\mathcal{X}} f(\nu_z)}{(1-\varepsilon)^2(k-2)}.$$

*The space required by the sketch at any given time is $O(k)$ in expectation. Additionally, with probability at least $1 - \delta$, the space will not exceed $O\left(k + \min\{\log m, \log\log\left(\frac{\mathsf{Sum}_D}{\mathsf{Min}(D)}\right)\} + \log\left(\frac{1}{\delta}\right)\right)$ at any time while processing $D$, where $m$ is the number of elements in $D$, $\mathsf{Min}(D)$ is the minimum value of an element in $D$, and $\mathsf{Sum}_D$ is the sum of frequencies of all keys.*

**Remark 5.2.** *The parameter $\varepsilon$ mainly affects the run time of processing an element. For each element processed by the stream, we generate $r = \frac{k}{\varepsilon}$ output elements that are then further processed by the sketch. Hence, the run time of processing an element grows with $\frac{1}{\varepsilon}$. The space is affected by $\varepsilon$ when considering worst case over the randomness. The total number of possible keys for output elements is $r$ times the number of active keys, and in the worst case (over the randomness), we may store all of them in* Sideline.

The sketch and estimator specification use the following functions in a black-box fashion

$$A(\gamma) \quad := \quad \int_\gamma^\infty a(t)dt$$

$$B(\gamma) \quad := \quad \int_0^\gamma ta(t)dt$$

where $a(t)$ is the inverse complement Laplace transform of $f$, as specified in Equation (6) (Section 2.4). Closed expressions for $A(t)$ and $B(t)$ for some common concave sublinear functions $f$ are provided in [10]. These functions are well-defined for any soft concave sublinear $f$. Also note that it suffices to approximate the values of $A$ and $B$ within a small multiplicative error (which will carry over to the variance, see Remark 5.11), so one can also use a table of values to compute the function.

While processing the stream, we will keep track of the sum of values of all elements $\mathsf{Sum}_D = \sum_{x\in\mathcal{X}} \nu_x$. We will then use $\mathsf{Sum}_D$ to set $\gamma$ adaptively to be $\frac{2\varepsilon}{\mathsf{Sum}_D}$. Thus, this is a running "candidate" value that can only decrease over time. The final value of $\gamma$ will be set when we produce a sample from the sketch in Algorithm 5. In the discussion below, we will show that setting $\gamma = \frac{2\varepsilon}{\mathsf{Sum}_D}$ satisfies the conditions needed for each of the sketches for $\mathcal{L}^c[a](\nu_x)_0^\gamma$ and $\mathcal{L}^c[a](\nu_x)_\gamma^\infty$.

## 5.1 The Sketch for $\mathcal{L}^c[a](\nu_x)_0^\gamma$

For the contributions $\mathcal{L}^c[a](\nu_x)_0^\gamma = \int_0^\gamma a(t)(1 - e^{-\nu_x t})dt$, we will see that as long as we choose a small enough $\gamma$, $\int_0^\gamma a(t)(1 - e^{-\nu_x t})dt$ will be approximately $\left(\int_0^\gamma a(t)tdt\right)\nu_x$, up to a multiplicative

$1 - \varepsilon$ factor. Note that $\left(\int_0^\gamma a(t)tdt\right)\nu_x$ is simply the frequency $\nu_x$ scaled by $B(\gamma) = \int_0^\gamma a(t)tdt$. A PPSWOR sample is invariant to the scaling, so we can simply use a PPSWOR sampling sketch according to the frequencies $\nu_x$ (Algorithm 2). The scaling only needs to be considered in a final step when the samples of the two sets of contributions are combined to produce a single sample.[7]

**Lemma 5.3.** *Let $\varepsilon > 0$ and $\gamma \leq \frac{2\varepsilon}{\max_x \nu_x}$. Then, for any key $x$,*

$$(1 - \varepsilon)\left(\int_0^\gamma a(t)tdt\right)\nu_x \leq \int_0^\gamma a(t)(1 - e^{-\nu_x t})dt \leq \left(\int_0^\gamma a(t)tdt\right)\nu_x.$$

*Proof.* Consider a key $x$ with frequency $\nu_x$. Using $1 - e^{-z} \leq z$, we get

$$\int_0^\gamma a(t)(1 - e^{-\nu_x t})dt \leq \int_0^\gamma a(t)\nu_x tdt = \left(\int_0^\gamma a(t)tdt\right)\nu_x.$$

Now, using $1 - e^{-z} \geq z - \frac{z^2}{2}$ for $z \geq 0$,

$$\int_0^\gamma a(t)(1 - e^{-\nu_x t})dt \geq \int_0^\gamma a(t)\left(\nu_x t - \frac{(\nu_x t)^2}{2}\right)dt.$$

Note that $\gamma \leq \frac{2\varepsilon}{\max_y \nu_y} \leq \frac{2\varepsilon}{\nu_x}$. Hence, for every $0 \leq t \leq \gamma$, $\nu_x t \leq 2\varepsilon$, and $\nu_x t - \frac{(\nu_x t)^2}{2} \geq (1 - \varepsilon)\nu_x t$. As a result, we get that

$$\int_0^\gamma a(t)(1 - e^{-\nu_x t})dt \geq (1 - \varepsilon)\left(\int_0^\gamma a(t)tdt\right)\nu_x.$$

$\square$

Note that our choice of $\gamma = \frac{2\varepsilon}{\mathsf{Sum}_D}$ satisfies the condition of Lemma 5.3.

## 5.2 The Sketch for $\mathcal{L}^c[a](\nu_x)_\gamma^\infty$

The sketch for $\mathcal{L}^c[a](\nu_x)_\gamma^\infty = \int_\gamma^\infty a(t)(1 - e^{-\nu_x t})dt$ processes elements in the following way. We map each input element $e = (e.key, e.val)$ into $r = \frac{k}{\varepsilon}$ output elements with keys $(e.key, 1)$ through $(e.key, r)$ and values $Y_i \sim \mathsf{Exp}[e.val]$ drawn independently. Each of these elements is then processed separately.

The main component of the sketch is a $\mathsf{SumMax}$ sampling sketch of size $k$. Our goal is that for each generated output element $((e.key, i), Y_i)$, the $\mathsf{SumMax}$ sketch will process an element $((e.key, i), A(\max\{Y_i, \gamma\}))$. However, since $\gamma$ decreases over time and we do not know its final value, we only process the elements with $Y_i \geq \gamma$ into the $\mathsf{SumMax}$ sketch. We keep the rest of the elements in an auxiliary structure (implemented as a maximum priority queue) that we call $\mathsf{Sideline}$. Every time we update the value of $\gamma$, we remove the elements with $Y_i \geq \gamma$ and process them into the $\mathsf{SumMax}$ sketch. Thus, at any time the $\mathsf{Sideline}$ structure only contains elements with value less than $\gamma$.[8]

For any active input key $x$ and $i \in [r]$, let $M_{x,i}$ denote the minimum value $Y_i$ that was generated with key $(x, i)$. We have the following invariants that we will use in our analysis:

1. Either the element $((x, i), M_{x,i})$ is in $\mathsf{Sideline}$ or an element $((x, i), A(M_{x,i}))$ was processed by the $\mathsf{SumMax}$ sketch.

2. Since $\gamma$ is decreasing over time, all elements ejected from the $\mathsf{Sideline}$ have value that is at least $\gamma$.

We also will use the following property of the sketch.

**Lemma 5.4.** *For any key $x$ that was active in the input and $i \in [r]$, $M_{x,i} \sim \textsf{Exp}[\nu_x]$ and these random variables are independent for different pairs $(x, i)$.*

*Proof.* $M_{x,i}$ by definition is the minimum of independent exponential random variables with sum of parameters $\nu_x$. $\qquad\square$

In the following lemma, we bound the size of $\textsf{Sideline}$ (and as a result, the entire sketch) with probability $1 - \delta$ for $0 < \delta < 1$ of our choice.

**Lemma 5.5.** *For a set of elements $D$, denote by $m$ the number of elements in $D$, and let $\textsf{Min}(D)$ denote the minimum value of any element in $D$. The expected number of elements in $\textsf{Sideline}$ at any given time is $O(k)$, and with probability at least $1 - \delta$, the number of elements in $\textsf{Sideline}$ will not exceed $O\left(k + \min\{\log m, \log\log\left(\frac{Sum_D}{Min(D)}\right)\} + \log\left(\frac{1}{\delta}\right)\right)$ at any time while processing $D$.*

The proof is deferred to Appendix D.

### 5.3 Generating a Sample from the Sketch

The final sample returned by Algorithm 5 is the merge of two samples:

1. The PPSWOR sample for $\mathcal{L}^c[a](\nu_x)_0^\gamma$ with frequencies scaled by $B(\gamma) = \int_0^\gamma ta(t)dt$.

2. The $\textsf{SumMax}$ sample for $\mathcal{L}^c[a](\nu_x)_\gamma^\infty$ with weights scaled by $\frac{1}{r}$. Before the scaling, the $\textsf{SumMax}$ sample processes an element $(e.key, A(\gamma))$ for each remaining $e \in \textsf{Sideline}$.

The scaling is performed using a property of exponential random variables and is formalized in the following lemma.

**Lemma 5.6.** *Given a PPSWOR sample where each key $x$ has frequency $\nu_x$, we can obtain a PPSWOR sample for the weights $c \cdot \nu_x$ by returning the original sample of keys but dividing the seed value of each key by $c$.*

*Proof.* A property of exponential random variables is that if $Y \sim \textsf{Exp}[w]$, then for any constant $c > 0$, $y/c \sim \textsf{Exp}[cw]$. Consider the set of seed values $\{\texttt{seed}(x) \mid x \in \mathcal{X}\}$ computed for the original PPSWOR sample according to the frequencies $\nu_x$. If we divided each seed value by $c$, the seed of key $x$ would come from the distribution $\textsf{Exp}(c\nu_x)$. Hence, a PPSWOR sample according to the weights $c\nu_x$ would contain the $k$ keys with lowest seed values after dividing by $c$, and these $k$ keys are the same keys that have lowest seed values before dividing by $c$. $\qquad\square$

Denote by $E$ the set of all elements that are passed on to the $\textsf{SumMax}$ sketch, either during the processing of the set of elements $D$ or in the final phase.

**Lemma 5.7.** *The final sample computed by Algorithm 5 is a PPSWOR sample with respect to weights*

$$V_x = \frac{1}{r} \, \textsf{SumMax}_E(x) + \nu_x \int_0^\gamma ta(t)dt \; .$$

*Proof.* The sample $s.ppswor$ in Algorithm 5 is a PPSWOR sample with respect to frequencies $\nu_x$, which is then scaled by $\int_0^\gamma ta(t)dt$ to get a PPSWOR sample according to the weights $\nu_x \int_0^\gamma ta(t)dt$. The sample $s.\textsf{SumMax}$ is a $\textsf{SumMax}$ sample, which by Corollary 4.2 is a PPSWOR sample according to the weights $\textsf{SumMax}_E(x)$. This sample is scaled to be a PPSWOR sample according to the weights $\frac{1}{r}\textsf{SumMax}_E(x)$.

Note that these samples are independent. When we perform a bottom-$k$ merge of the two samples, the seed of key $x$ is then the minimum of two independent exponential random variables with parameters $\int_0^\gamma ta(t)dt$ and $\frac{1}{r}\textsf{SumMax}_E(x)$. Therefore, the distribution of $\texttt{seed}(x)$ in the merged sample is $\textsf{Exp}(\int_0^\gamma ta(t)dt + \frac{1}{r}\textsf{SumMax}_E(x))$, which means that the sample is a PPSWOR sample according to those weights, as desired. $\qquad\square$

We next interpret $\frac{1}{r} \mathsf{SumMax}_E(x)$. From the invariants listed in Section 5.2 and the description of Algorithm 5, we have that for any active input key $x$ and $i \in [r]$, the element $((x, i), A(\max\{\gamma, M_{x,i}\}))$ was processed by the $\mathsf{SumMax}$ sketch. Because $A$ is monotonically non-increasing,

$$\mathsf{Max}_E((x, i)) = \max_{e \in E \mid e.key=(x,i)} e.val = A(\max\{\gamma, M_{x,i}\}).$$

Now, $\frac{1}{r} \mathsf{SumMax}_E(x) = \sum_{i=1}^{r} \frac{1}{r} \mathsf{Max}_E((x, i))$. By Lemma 5.4, the summands $\frac{1}{r} \mathsf{Max}_E((x, i))$ are independent random variables for every $x$ and $i$. By the monotonicity of the function $A$, each summand $\frac{1}{r} \mathsf{Max}_E((x, i))$ is in the range $\left[0, \frac{A(\gamma)}{r}\right]$.

The final sample returned by Algorithm 5 is then a stochastic PPSWOR sample as defined in Section 3. The weight of key $x$ also includes the deterministic component $\nu_x \int_0^\gamma t a(t) dt$, which can be larger than $\frac{A(\gamma)}{r}$. However, since this is summand is deterministic, we can break it into smaller deterministic parts, each of which will be at most $\frac{A(\gamma)}{r}$. This way, the sample still satisfies the condition that the weight of every key is the sum of independent random variables in $\left[0, \frac{A(\gamma)}{r}\right]$. The next step is to show that it satisfies the conditions of Theorem 3.1.

**Lemma 5.8.** *For a key $x$, define $V_x = \frac{1}{r} \mathsf{SumMax}_E(x) + \nu_x \int_0^\gamma t a(t) dt$. Let $V = \sum_{x \in \mathcal{X}} V_x$. Then, for any $0 < \varepsilon \le \frac{1}{2}$ and $r \ge \frac{k}{\varepsilon}$,*

$$E[V] \ge \frac{A(\gamma)}{r} \cdot k.$$

The proof, which is deferred to Appendix D, uses the following lemma which is due to Cohen [10].

**Lemma 5.9.** *[10] For every input key $x$ and $i = 1, \dots, r$,*

$$E[\mathsf{Max}_E((x, i))] = \mathcal{L}^c[a](\nu_x)_\gamma^\infty = \int_\gamma^\infty a(t)(1 - e^{-\nu_x t}) dt.$$

The following theorem combines the previous lemmas to show how to estimate $f(\nu_x)$ using the sample and bound the variance. Note that we need to specify how to compute the estimator. For getting $f(\nu_x)$ we make another pass, and the computation of the conditioned inclusion probability is described in the next subsection.

**Theorem 5.10.** *The sample returned by Algorithm 5 is a stochastic PPSWOR sample, where each key $x$ has weight $V_x$ that satisfies $f(\nu_x) \le E[V_x] \le \frac{1}{(1-\varepsilon)} f(\nu_x)$. The per-key inverse-probability estimator according to the weights $f(\nu_x)$,*

$$\widehat{f(\nu_x)} = \begin{cases} \frac{f(\nu_x)}{\Pr[\mathsf{seed}(x) < \tau]} & x \in S \\ 0 & x \notin S \end{cases}.$$

*is unbiased and has variance*

$$\mathsf{Var}\left[\widehat{f(\nu_x)}\right] \le \frac{4E[V_x]E[V]}{k-2} \le \frac{4f(\nu_x) \sum_{z \in \mathcal{X}} f(\nu_z)}{(1-\varepsilon)^2(k-2)}$$

*where $V = \sum_{x \in \mathcal{X}} V_x$.*

*Proof.* We first prove that $f(\nu_x) \le E[V_x] \le \frac{1}{(1-\varepsilon)} f(\nu_x)$. The randomized weight $V_x$ is the sum of two terms:

$$V_x = \frac{1}{r} \mathsf{SumMax}_E(x) + \nu_x \int_0^\gamma t a(t) dt.$$

As in the proof of Lemma 5.8, using Lemma 5.9,

$$E\left[\frac{1}{r} \mathsf{SumMax}_E(x)\right] = \frac{1}{r} \sum_{i=1}^{r} E[\mathsf{Max}_E((x, i))] = \int_\gamma^\infty a(t)(1 - e^{-\nu_x t}) dt. \qquad (8)$$

The quantity $\nu_x \int_0^\gamma t a(t) dt$ is deterministic, and since $\gamma \le \frac{2\varepsilon}{\max_x \nu_x}$, by Lemma 5.3,

$$\int_0^\gamma a(t)(1 - e^{-\nu_x t}) dt \le \nu_x \int_0^\gamma t a(t) dt \le \frac{1}{1-\varepsilon} \int_0^\gamma a(t)(1 - e^{-\nu_x t}) dt. \qquad (9)$$

Recall that $f(\nu_x) = \int_0^\infty a(t)(1 - e^{-\nu_x t})dt = \int_0^\gamma a(t)(1 - e^{-\nu_x t})dt + \int_\gamma^\infty a(t)(1 - e^{-\nu_x t})dt$. Combining Equations (8) and (9), we get that

$$f(\nu_x) \le \mathsf{E}[V_x] \le \frac{1}{(1-\varepsilon)}f(\nu_x).$$

Now consider the estimator $\widehat{f(\nu_x)}$ for $f(\nu_x)$. To show that the estimator is unbiased, that is, $\mathsf{E}\left[\widehat{f(\nu_x)}\right] = f(\nu_x)$, we can follow the proof of Claim 2.4 exactly as written earlier. It is left to bound the variance. For the sake of the analysis, consider the following estimator for $\mathsf{E}[V_x]$:

$$\widehat{\mathsf{E}[V_x]} = \begin{cases} \frac{\mathsf{E}[V_x]}{\Pr[\mathtt{seed}(x) < \tau]} & x \in S \\ 0 & x \notin S \end{cases}.$$

This is again the inverse-probability estimator from Definition 2.3. Our sample is a stochastic PPSWOR sample according to the weights $V_x$, where each one of $V_x$ is a sum of independent random variables in $\left[0, \frac{A(\gamma)}{r}\right]$ (recall that the deterministic part can also be expressed as a sum with each summand in $\left[0, \frac{A(\gamma)}{r}\right]$). Lemma 5.8 shows that $\mathsf{E}[V] \ge \frac{A(\gamma)}{r} \cdot k$. Hence, we satisfy the conditions of Theorem 3.1, which in turn shows that

$$\mathsf{Var}\left[\widehat{\mathsf{E}[V_x]}\right] \le \frac{4\mathsf{E}[V_x]\mathsf{E}[V]}{k - 2}.$$

Finally, note that $\widehat{f(\nu_x)} = \frac{f(\nu_x)}{\mathsf{E}[V_x]} \cdot \widehat{\mathsf{E}[V_x]}$. We established above that $\frac{f(\nu_x)}{\mathsf{E}[V_x]} \le 1$ and $\mathsf{E}[V_x] \le \frac{1}{(1-\varepsilon)}f(\nu_x)$. We conclude that

$$\begin{aligned}
\mathsf{Var}\left[\widehat{f(\nu_x)}\right] &= \mathsf{Var}\left[\frac{f(\nu_x)}{\mathsf{E}[V_x]} \cdot \widehat{\mathsf{E}[V_x]}\right] \\
&= \left(\frac{f(\nu_x)}{\mathsf{E}[V_x]}\right)^2 \mathsf{Var}\left[\widehat{\mathsf{E}[V_x]}\right] \\
&\le \frac{4\mathsf{E}[V_x]\mathsf{E}[V]}{k - 2} \\
&\le \frac{4f(\nu_x)\sum_{z \in \mathcal{X}} f(\nu_z)}{(1-\varepsilon)^2(k-2)}.
\end{aligned}$$

$\square$

**Remark 5.11.** *The theorem establishes a bound on the variance of the estimator for $\mathsf{E}[V_x]$, and then uses it to bound the variance of the estimator for $f(\nu_x)$, which is possible since $\mathsf{E}[V_x]$ approximates $f(\nu_x)$. This results in increasing the variance by an $\varepsilon$-dependent factor. Similarly, if we wish to estimate $f(\nu_x)$ for a concave sublinear function $f$ (and not a* soft *concave sublinear function), we can use the same idea and lose another constant factor in the variance.*

### 5.4 Expressing Conditioned Inclusion Probabilities

To compute the estimator $\widehat{f(\nu_x)}$, we need to know both $f(\nu_x)$ and the precise conditioned inclusion probability $\Pr[\mathtt{seed}(x) < \tau]$. In order to get $f(\nu_x)$, we perform a second pass over the data elements to obtain the exact frequencies $\nu_x$ for $x \in S$. This can be done via a simple composable sketch that collects and sums the values of data elements with keys that occur in the sample $S$.

We next consider computing the conditioned inclusion probabilities. The following lemma considers the $\mathtt{seed}$ distributions of keys in the final sample. It shows that the distributions are parameterized by $\nu_x$ and describes their CDF.

**Lemma 5.12.** *Algorithms 4 and 5 describe a bottom-$k$ sampling scheme, where in the output sample the seed of each key $x$ is drawn from a distribution $\mathsf{SeedDist}^{(F)}[\nu_x]$. The distribution $\mathsf{SeedDist}^{(F)}[w]$ has the following cumulative distribution function:*

$$\mathsf{SeedCDF}^{(F)}(w, t) := \Pr_{s \sim \mathsf{SeedDist}^{(F)}[w]}[s < t] = 1 - p_1 p_2^r,$$

*where*

$$\begin{aligned}
p_1 &= \exp(-wB(\gamma)t) \\
p_2 &= \int_0^\infty w \exp(-wy) \exp(-A(\max\{y,\gamma\})t/r)dy
\end{aligned}$$

The proof is deferred to Appendix D.

## 6  Experiments

We implemented our sampling sketch and report here the results of experiments on real and synthetic datasets. Our experiments are small-scale and aimed to demonstrate the simplicity and practicality of our sketch design and to understand the actual space and error bounds (that can be significantly better than our worst-case bounds).

### 6.1  Implementation

Our Python 2.7 implementation follows the pseudocode of the sampling sketch (Algorithm 4), the PPSWOR (Algorithm 2) and SumMax (Algorithm 3) substructures, the sample production from the sketch (Algorithm 5), and the estimator (that evaluates the conditioned inclusion probabilities, see Section 5.4). We incorporated two practical optimizations that are not shown in the pseudocode. These optimizations do not affect the outcome of the computation or the worst-case analysis, but reduce the sketch size in practice.

**Removing redundant keys from the PPSWOR subsketch**  The pseudocode (Algorithm 4) maintains two samples of size $k$, the PPSWOR and the SumMax samples. The final sample of size $k$ is obtained by merging these two samples. Our implementation instead maintains a truncated PPSWOR sketch that removes elements that are already redundant (do not have a potential to be included in the merged sample). We keep an element in the PPSWOR sketch only when the seed value is lower than $rB(\gamma)$ times the current threshold $\tau$ of the SumMax sample. This means that the "effective" inclusion threshold we use for the PPSWOR sketch is the minimum of the $k$th largest (the threshold of the PPSWOR sketch) and $rB(\gamma)\tau$. To establish that elements that do not satisfy this condition are indeed redundant, recall that when we later merge the PPSWOR and the SumMax samples, the value of $B(\gamma)$ can only become lower and the SumMax threshold can only be lower, making inclusion more restrictive. This optimization may result in maintaining much fewer than $k$ elements and possibly an empty PPSWOR sketch. The benefit is larger for functions when $A(t)$ is bounded (as $t$ approaches 0). In particular, when $a(t) = 0$ for $t \leq \gamma$ we get $B(\gamma) = 0$ and the truncation will result in an empty sample.

**Removing redundant elements from Sideline**  The pseudocode may place elements in Sideline that have no future potential of modifying the SumMax sketch. In our implementation, we place and keep an element $((e.key, i), Y)$ in Sideline only as long as the following condition holds: If $((e.key, i), A(Y))$ is processed by the current SumMax sketch, it would modify the sketch. To establish redundancy of discarded elements, note that when an element is eventually processed, the value it is processed with is at most $A(Y)$ (can be $A(\gamma)$ for $\gamma \geq Y$) and also at that point the SumMax sketch threshold can only be more restrictive.

### 6.2  Datasets and Experimental Results

We used the following datasets for the experiments:

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

# Acknowledgments

Ofir Geri was supported by NSF grant CCF-1617577, a Simons Investigator Award for Moses Charikar, and the Google Graduate Fellowship in Computer Science in the School of Engineering at Stanford University. The computing for this project was performed on the Sherlock cluster. We would like to thank Stanford University and the Stanford Research Computing Center for providing computational resources and support that contributed to these research results.

## Footnotes

[1]For streaming algorithms we are typically interested in deterministic worst-case bounds on the space, but streaming algorithms with randomized space have also been considered in some cases, in particular when studying the sliding window model [3, 7].

[2]In the general case, we assume these functions of the frequency are computationally tractable or can be easily approximated up to a small constant factor. For our application, the discussion will follow in Section 5.4.

[3]Defined as the ratio of the standard deviation to mean. For our unbiased estimators it is equal to the relative root mean squared error.

[4]We note in our applications, lower values mean higher inclusion probabilities. In most applications, higher values are associated with better results, and accordingly, *first-order stochastic dominance* is usually defined as the reverse.

[5]The definition also allows $a(t)$ to have discrete mass at points (that is, we can add a component of the form $\sum_i a(t_i)(1 - e^{-\nu t_i})$). We generally ignore this component for the sake of presentation, but one way to model this is using Dirac delta.

[6]Here we also allow $a(t)$ to have discrete mass using Dirac delta. For the sake of presentation, we also assume bounded $\nu$ – otherwise we need to add a linear component $A_\infty \nu$ for some $A_\infty \geq 0$. The component $A_\infty \nu$ can easily be added to the final sketch presented in Section 5, for example, by taking the minimum with another independent PPSWOR sketch.

[7]In our implementation (Section 6) we incorporated an optimization where we only keep in the PPSWOR sample elements that may contribute to the final sample.

[8]In our implementation (see Section 6) we only keep in $\mathsf{Sideline}$ elements that have the potential to modify the SumMax sketch when inserted.

[9] $X$ takes values in $\mathbb{R}$ and cannot be $\infty$. However, the random variable $X$ represents the minimum seed value (or the inclusion threshold $\tau_x$ as in Section 2.3), and the event $X \leq \tau$ represents whether the inclusion threshold for key $x$ is at most $\tau$. The case $S = 0$ corresponds to no elements generated with keys in $\mathcal{X} \setminus \{x\}$, so we can say that the inclusion threshold for $x$ is $\infty$ (the event we care about is whether $x$ enters the sample or not). Here we are trying to show a distribution that dominates the distribution of the inclusion threshold, and for that purpose, any threshold $\tau > 0$ is more restrictive than $\infty$. From a technical perspective, when $S = 0$, we can still use the CDF of $\textsf{Exp}[S]$ since $\Pr[X \leq \tau] = 0 = 1 - e^{-S\tau}$. Later, when we consider the $k$-th lowest seed, we will similarly allow it to be $\infty$ when less than $k$ keys are active.

[10] It may be the case that $\gamma$ is never $\frac{1}{2}t$, but in that case we consider the minimum value of $\gamma$ that is at least $\frac{1}{2}t$.

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

## A  Proofs Deferred from Section 2

*Proof of Proposition 2.1.* Each data element $e = (e.key, e.val)$ is processed by giving it a score $\texttt{ElementScore}(e) \sim \textsf{Exp}(e.val)$ and then processing the element $(e.key, \texttt{ElementScore}(e))$ by the bottom-$k$ structure. For a key $x$, we define

$$\texttt{seed}(x) := \min_{e \in D | e.key = x} \texttt{ElementScore}(e)$$

to be the smallest score of an element with key $x$.

Since $\texttt{seed}(x)$ is the minimum of independent exponential random variables, its distribution is $\textsf{Exp}(w_x)$. After processing the elements in $D$, the bottom-$k$ structure contains the $k$ pairs $(x, \texttt{seed}(x))$ with smallest $\texttt{seed}(x)$ values, and hence obtains the respective PPSWOR sample.

Consider now the sketch resulting from merging two sampling sketches computed for $D_1$ and $D_2$. For each key $x$, denote by $\texttt{seed}_1(x)$ and $\texttt{seed}_2(x)$ the values of $x$ in the sketches for $D_1$ and $D_2$, respectively. Then, the merged sketch contains the $k$ pairs $(x, \texttt{seed}(x))$ with smallest $\texttt{seed}(x) := \min\{seed_1(x), seed_2(x)\}$ values. As $\texttt{seed}(x)$ is the minimum of two independent exponential random variables with parameters $\textsf{Sum}_{D_1}(x)$ and $\textsf{Sum}_{D_2}(x)$, we get that $\texttt{seed}(x) \sim \textsf{Exp}(\textsf{Sum}_{D_1 \cup D_2}(x))$, as desired. $\square$

# B Proofs Deferred from Section 3

## B.1 Threshold Distribution and Fixed-Threshold Inclusion Probability

Before proceeding, we establish two technical lemmas that will be useful later. The first lemma shows that the distribution of the $k$-th lowest seed is dominated by the Erlang distribution which takes the sum of the expected weights $V$ as a parameter. The lemma will be useful later when we consider the inclusion threshold $\tau_x$.

**Lemma B.1.** *Consider a set of keys $\mathcal{X}$, such that the weight of each $x \in \mathcal{X}$ is a random variable $S_x \geq 0$. Let $V = \sum_{x \in \mathcal{X}} \mathsf{E}[S_x]$, and for each $x$ with $S_x > 0$, draw $\texttt{seed}(x) \sim \textsf{Exp}[S_x]$ independently. Then, the distribution of the $k$-th lowest seed, $\{\texttt{seed}(x) \mid x \in X\}_{(k)}$, is dominated by $\textsf{Erlang}[V, k]$.*

We first establish the dominance relation $\textsf{Exp}[S] \preceq \textsf{Exp}[\mathsf{E}[S]]$ for any nonnegative random variable $S$.

**Lemma B.2.** *Let $S \geq 0$ be a random variable. Let $X$ be a random variable such that $X \sim \textsf{Exp}[S]$ when $S > 0$ and $X = \infty$ otherwise.[9] Then, the distribution of $X$ is dominated by $\textsf{Exp}[\mathsf{E}[S]]$, that is, $\forall \tau, \Pr_{X \sim \textsf{Exp}[S]}[X \leq \tau] \leq 1 - e^{-\mathsf{E}[S]\tau}$.*

*Proof.* Follows from Jensen's inequality:
$$\Pr[X \leq \tau] = \mathsf{E}_S[1 - e^{-S\tau}] = 1 - \mathsf{E}_S[e^{-S\tau}] \leq 1 - e^{-\mathsf{E}[S]\tau}$$
$\square$

Therefore, for any key $x$, the $\texttt{seed}(x)$ distribution with stochastic weights is dominated by $\textsf{Exp}[\mathsf{E}[S_x]]$, which is the distribution used by PPSWOR according to the expected weights. We now consider the distribution of $\{\texttt{seed}(x) \mid x \in X\}_{(k)}$, which is the $k$-th lowest seed value. We show that the distribution of the $k$-th lowest seed is dominated by $\textsf{Erlang}[V, k]$ (recall that $V = \sum_x \mathsf{E}[S_x]$). In the proof, we will use the following property of dominance (the proof of the following property is standard and included here for completeness).

**Claim B.3.** *Let $X_1, \ldots, X_k, Y_1, \ldots, Y_k$ be independent random variables such that the distribution of $X_i$ is dominated by that of $Y_i$. Then the distribution of $X_1 + \ldots + X_k$ is dominated by that of $Y_1 + \ldots + Y_k$.*

*Proof.* We prove for $k = 2$ (the proof for $k > 2$ follows from a simple induction argument). Denote by $f_i$ and $F_i$ the PDF and CDF functions of $X_i$, respectively, and by $g_i$ and $G_i$ the PDF and CDF of $Y_i$. From the dominance assumption, we know that $F_i(t) \leq G_i(t)$ for all $i$. Now,

$$\Pr[X_1 + X_2 < t] = \int_0^\infty f_1(x) \Pr[X_2 < t - x] dx$$
$$= \int_0^t f_1(x) F_2(t - x) dx$$
$$\leq \int_0^t g_1(x) F_2(t - x) dx$$
$$\text{[from dominance as } F_2(t - x) \text{ is non-increasing in } x]$$
$$\leq \int_0^t g_1(x) G_2(t - x) dx$$
$$= \Pr[Y_1 + Y_2 < t]$$
$\square$

We are now ready to prove Lemma B.1.

*Proof of Lemma B.1.* Conditioned on the values of $S_x$, the distribution of $\{\texttt{seed}(x) \mid x \in X\}_{(k)}$ is dominated by $\mathsf{Erlang}[\sum_{x \in \mathcal{X}} S_x, k]$ (for a proof, see Appendix C in [11]). The distribution of $\{\texttt{seed}(x) \mid x \in X\}_{(k)}$ (unconditioned on the values of $S_x$) is a linear combination of distributions, which are each dominated by the respective Erlang distribution. Using the definition of dominance (Definition 2.6) and the law of total probability, we get that the distribution of $\{\texttt{seed}(x) \mid x \in X\}_{(k)}$ (unconditioned on $S_x$) is dominated by $\mathsf{Erlang}[\sum_{x \in \mathcal{X}} S_x, k]$ (unconditioned on $S_x$). A random variable drawn from $\mathsf{Erlang}[\sum_{x \in \mathcal{X}} S_x, k]$ has the same distribution as the sum of $k$ independent random variables drawn from $\mathsf{Exp}(\sum_{x \in \mathcal{X}} S_x)$. The distribution of each of these $k$ exponential random variables is dominated by $\mathsf{Exp}(V)$ (by Lemma B.2). Using Claim B.3, we get that $\mathsf{Erlang}[\sum_{x \in \mathcal{X}} S_x, k] \preceq \mathsf{Erlang}[V, k]$. The assertion of the lemma then follows from the transitivity of dominance. $\qquad\square$

The second lemma provides lower bounds on the CDF of $\mathsf{Exp}(S)$ under certain conditions.

**Lemma B.4.** *Let the random variable $S = \sum_{i=1}^{r} S_i$ be a sum of $r$ independent random variables in the range $[0, T]$. Let $v = \mathsf{E}[S]$. Then,*

$$\Pr_{X \sim \mathsf{Exp}[S]}[X \leq \tau] \geq 1 - e^{-v\tau(1 - \tau T/2)} \, .$$

In the regime $\tau T < 1$, we get that the probability of being less than $\tau$ is close to that of $\mathsf{Exp}(\mathsf{E}[S])$.

**Lemma B.5.** *Let $S$ be a random variables in $[0, T]$ with expectation $\mathsf{E}[S] = v$. Then for all $\tau$,*

$$\Pr_{X \sim \mathsf{Exp}[S]}[X \leq \tau] \geq \frac{v}{T}(1 - e^{-T\tau}) \, .$$

*Proof.* Denote the probability density function of $S$ by $p_S$. Conditioned on the value of $S$, the probability of $X \sim \mathsf{Exp}[S]$ being below $\tau$ is

$$\Pr[X \leq \tau \mid S = s] = 1 - e^{-s\tau}.$$

It follows that

$$\Pr[X \leq \tau] = \mathsf{E}[1 - e^{-S\tau}] = \int_0^T p_S(x)(1 - e^{-x\tau}) dx.$$

Consider the function $f(x) = 1 - e^{-x\tau}$ for a fixed $\tau \geq 0$. Since $f$ is concave, for every $x \in [0, T]$,

$$\begin{aligned}
f(x) &= f\left(\left(1 - \frac{x}{T}\right) \cdot 0 + \frac{x}{T} \cdot T\right) \\
&\geq \left(1 - \frac{x}{T}\right) \cdot f(0) + \frac{x}{T} \cdot f(T) \\
&= \frac{x}{T}(1 - e^{-T\tau}).
\end{aligned}$$

By monotonicity,

$$\int_0^T p_S(x)(1 - e^{-x\tau}) dx \geq \int_0^T p_S(x) \cdot \frac{x}{T}(1 - e^{-T\tau}) dx$$

and finally,

$$\Pr[X \leq \tau] \geq \frac{1 - e^{-T\tau}}{T} \cdot \int_0^T p_S(x) x \, dx = \frac{v}{T} \cdot (1 - e^{-T\tau}).$$

$\qquad\square$

**Lemma B.6.** *Let the random variable $S = \sum_{i=1}^{r} S_i$ be a sum of $r$ independent random variables in the range $[0, T]$. Let $v = \mathsf{E}[S]$. Then,*

$$\Pr_{X \sim \mathsf{Exp}[S]}[X \leq \tau] \geq 1 - \exp\left(-\frac{v}{T}(1 - e^{-T\tau})\right) \, .$$

*Proof.* Let $X \sim \mathsf{Exp}[S]$. Since $S = \sum_{i=1}^r S_i$, we could define $r$ independent exponential random variables $X_i \sim \mathsf{Exp}[S_i]$. $X$ has the same distribution as $\min_{1 \leq i \leq r} X_i$. Hence,

$$\Pr[X > \tau] = \Pr\left[\min_{1 \leq i \leq r} X_i > \tau\right]$$

$$= \prod_{i=1}^r \Pr\left[X_i > \tau\right]$$

$$\leq \prod_{i=1}^r \left(1 - \frac{\mathsf{E}[S_i]}{T} \cdot (1 - e^{-T\tau})\right)$$

$$\leq \left(1 - \frac{\mathsf{E}[S]}{rT} \cdot (1 - e^{-T\tau})\right)^r$$

where the last inequality follows from the arithmetic mean-geometric mean inequality. Now, using the inequality $1 - x \leq e^{-x}$ (for any $x \in \mathbb{R}$), and the fact that $f(x) = x^r$ is non-decreasing for $x, r \geq 0$, we get that

$$\Pr[X > \tau] \leq \exp\left(-\frac{\mathsf{E}[S]}{rT} \cdot (1 - e^{-T\tau}) \cdot r\right) = \exp\left(-\frac{\mathsf{E}[S]}{T} \cdot (1 - e^{-T\tau})\right).$$

Consequently,

$$\Pr[X \leq \tau] \geq 1 - \exp\left(-\frac{v}{T}(1 - e^{-T\tau})\right).$$

$\square$

*Proof of Lemma B.4.* Follows from Lemma B.6 using the inequality $1 - e^{-x} \geq x - x^2/2$ for $x \geq 0$. $\square$

## B.2   Variance Bounds for the Inverse-Probability Estimator

*Proof of Theorem 3.1.* We start bounding the per-key variance as in Claim 2.5:

$$\mathsf{Var}(\widehat{v_x}) = \mathsf{E}_{\tau_x}\left[v_x^2\left(\frac{1}{\Pr[\mathsf{seed}(x) < \tau_x]} - 1\right)\right].$$

By Lemma B.1, we know that the distribution of $\tau_x$ (the $k - 1$ lowest seed of the keys in $\mathcal{X} \setminus \{x\}$) is dominated by $\mathsf{Erlang}[V, k-1]$, hence

$$\mathsf{Var}(\widehat{v_x}) \leq \mathsf{E}_{t \sim \mathsf{Erlang}[V,k-1]}\left[v_x^2\left(\frac{1}{\Pr[\mathsf{seed}(x) < t]} - 1\right)\right]$$

$$= \int_0^\infty B_{V,k-1}(t) v_x^2\left(\frac{1}{\Pr[\mathsf{seed}(x) < t]} - 1\right) dt$$

$$= \int_0^{1/T} B_{V,k-1}(t) \cdot v_x^2\left(\frac{1}{\Pr[\mathsf{seed}(x) < t]} - 1\right) dt$$

$$+ \int_{1/T}^\infty B_{V,k-1}(t) \cdot v_x^2\left(\frac{1}{\Pr[\mathsf{seed}(x) < t]} - 1\right) dt$$

To bound the first summand, since $t \leq \frac{1}{T}$, we get from Lemma B.4 (applied to $\mathsf{seed}(x)$) that $\Pr[\mathsf{seed}(x) < t] \geq 1 - e^{-v_x t\left(1 - \frac{tT}{2}\right)} \geq 1 - e^{-v_x t/2}$. It follows that

$$\int_0^{1/T} B_{V,k-1}(t) \cdot v_x^2\left(\frac{1}{\Pr[\mathsf{seed}(x) < t]} - 1\right) dt$$

$$\leq \int_0^{1/T} B_{V,k-1}(t) \cdot v_x^2\left(\frac{1}{1 - e^{-v_x t/2}} - 1\right) dt$$

$$\leq \int_0^{1/T} B_{V,k-1}(t) \cdot \frac{v_x^2}{v_x t/2} dt \qquad \left[\frac{e^{-x}}{1 - e^{-x}} \leq \frac{1}{x}\right]$$

$$= 2 \int_0^{1/T} B_{V,k-1}(t) \cdot \frac{v_x}{t} dt$$

$$\leq 2 \int_0^{\infty} B_{V,k-1}(t) \cdot \frac{v_x}{t} dt$$

$$= \frac{2 v_x V}{k-2} \qquad \text{[PPSWOR analysis (Section 2.3)]}$$

To bound the second summand, since $t > \frac{1}{T}$, $\Pr[\texttt{seed}(x) < t] \geq \Pr[\texttt{seed}(x) < 1/T] \geq 1 - e^{-v_x/2T}$. Subsequently,

$$\int_{1/T}^{\infty} B_{V,k-1}(t) \cdot v_x^2 \left( \frac{1}{\Pr[\texttt{seed}(x) < t]} - 1 \right) dt$$

$$\leq \int_{1/T}^{\infty} B_{V,k-1}(t) \cdot v_x^2 \left( \frac{1}{1 - e^{-v_x/2T}} - 1 \right) dt$$

$$= v_x^2 \left( \frac{1}{1 - e^{-v_x/2T}} - 1 \right) \int_{1/T}^{\infty} B_{V,k-1}(t) dt$$

$$\leq v_x^2 \left( \frac{1}{1 - e^{-v_x/2T}} - 1 \right) \qquad \text{[integral of density]}$$

$$\leq \frac{v_x^2}{v_x/2T} \qquad [\frac{e^{-x}}{1 - e^{-x}} \leq \frac{1}{x}]$$

$$= 2 T v_x$$

$$\leq \frac{2 v_x V}{k} \qquad [V \geq Tk]$$

Combining, we get that

$$\mathsf{Var}\left(\widehat{v_x}\right) \leq \frac{2 v_x V}{k-2} + \frac{2 v_x V}{k} \leq \frac{4 v_x V}{k-2}.$$

$\square$

## B.3 Inclusion Probability in a Stochastic Sample

*Proof of Theorem 3.2.* We first separately deal with the case where there is only one key, which we denote $x$. In this case, $V = v_x$, and if $S_x > 0$, then $x$ is included in the sample. Otherwise, the sample is empty. In the proof of Lemma B.4, when $S = 0$, we used $\Pr[X \leq \tau] = 1 - e^{-s\tau} = 0$ and the event $X \leq \tau$ does not happen. Hence, we can use Lemma B.4 to bound $\Pr[S_x > 0] \geq \Pr[\texttt{seed}(x) \leq \tau]$ for any $\tau > 0$. We pick $\tau = \frac{2\varepsilon}{T}$ and get that $x$ is included in the sample with probability

$$\Pr[S_x > 0] \geq 1 - e^{-v \frac{2\varepsilon}{T}(1-\varepsilon)} \geq 1 - e^{-\frac{2\varepsilon}{\varepsilon}(1-\varepsilon) \ln\left(\frac{1}{\varepsilon}\right)} \geq 1 - \varepsilon$$

using $V \geq \frac{1}{\varepsilon} \ln\left(\frac{1}{\varepsilon}\right) T$ and $2(1-\varepsilon) \geq 1$.

If there is more than one key, a key $x$ is included in the sample if $\texttt{seed}(x)$ is smaller than the seed of all other keys. The distribution of $\min_{z \neq x} \texttt{seed}(z)$ is $\mathsf{Exp}\left(\sum_{z \neq x} S_z\right)$, which is dominated by $\mathsf{Exp}\left(\sum_{z \neq x} v_z\right)$ (Lemma B.2). Then,

$$\Pr[\texttt{seed}(x) < \min_{z \neq x} \texttt{seed}(z)]$$

$$\geq \mathsf{E}_{t \sim \mathsf{Exp}[V - v_x]} \Pr[\texttt{seed}(x) < t]$$

$$= \int_0^{\infty} (V - v_x) e^{-(V - v_x)t} \Pr[\texttt{seed}(x) < t] dt$$

$$\geq \int_0^{2\varepsilon/T} (V - v_x) e^{-(V - v_x)t} \Pr[\texttt{seed}(x) < t] dt$$

$$+ \int_{2\varepsilon/T}^{\infty} (V - v_x) e^{-(V - v_x)t} \Pr[\texttt{seed}(x) < 2\varepsilon/T] dt$$

$$\geq \int_0^{2\varepsilon/T} (V - v_x) e^{-(V-v_x)t} \left(1 - e^{-v_x t(1 - tT/2)}\right) dt$$

$$+ \int_{2\varepsilon/T}^{\infty} (V - v_x) e^{-(V-v_x)t} \left(1 - e^{-v_x \frac{2\varepsilon}{T}(1-\varepsilon)}\right) dt$$

$$\geq \int_0^{\infty} (V - v_x) e^{-(V-v_x)t} dt - \int_0^{2\varepsilon/T} (V - v_x) e^{-(V-v_x)t} e^{-v_x t(1-\varepsilon)} dt$$

$$- \int_{2\varepsilon/T}^{\infty} (V - v_x) e^{-(V-v_x)t} e^{-v_x \frac{2\varepsilon}{T}(1-\varepsilon)} dt$$

$$= 1 - \int_0^{2\varepsilon/T} (V - v_x) e^{-(V-\varepsilon v_x)t} dt - e^{-(V-v_x)\frac{2\varepsilon}{T}} e^{-v_x \frac{2\varepsilon}{T}(1-\varepsilon)}$$

$$= 1 - \frac{V - v_x}{V - \varepsilon v_x} \int_0^{2\varepsilon/T} (V - \varepsilon v_x) e^{-(V-\varepsilon v_x)t} dt - e^{-(V-\varepsilon v_x)\frac{2\varepsilon}{T}}$$

$$= 1 - \frac{V - v_x}{V - \varepsilon v_x} \left(1 - e^{-(V-\varepsilon v_x)\frac{2\varepsilon}{T}}\right) - e^{-(V-\varepsilon v_x)\frac{2\varepsilon}{T}}$$

$$= \left(1 - \frac{V - v_x}{V - \varepsilon v_x}\right) \left(1 - e^{-(V-\varepsilon v_x)\frac{2\varepsilon}{T}}\right)$$

$$= \frac{(1-\varepsilon)v_x}{V - \varepsilon v_x} \left(1 - e^{-(V-\varepsilon v_x)\frac{2\varepsilon}{T}}\right)$$

$$\geq \frac{(1-\varepsilon)v_x}{V} \left(1 - e^{-\frac{2\varepsilon}{T}(1-\varepsilon)V}\right) \qquad [V \geq v_x]$$

$$\geq \frac{(1-\varepsilon)v_x}{V} \left(1 - e^{-\frac{2\varepsilon(1-\varepsilon)}{\epsilon} \ln\left(\frac{1}{\varepsilon}\right)}\right)$$

$$\geq \frac{(1-\varepsilon)v_x}{V} \left(1 - e^{-\ln\left(\frac{1}{\varepsilon}\right)}\right) \qquad [2(1-\varepsilon) \geq 1]$$

$$= (1-\varepsilon)^2 \cdot \frac{v_x}{V}$$

$$\geq (1 - 2\varepsilon)\frac{v_x}{V}.$$

$\square$

## C  Proofs Deferred from Section 4

*Proof of Lemma 4.1.*  With a slight abuse of notation, for a full key $z = (z.p, z.s)$ we define

$$\texttt{seed}_D(z) := \min_{e \in D | e.key = z} \texttt{ElementScore}(e).$$

Now, since we use the same value $h(z)$ for all elements with key $z$, the minimum $\texttt{ElementScore}(e)$ value generated for an element $e \in D$ with key $e.key = z$ is $h(z)/\,\mathsf{Max}_D(z)$:

$$\texttt{seed}_D(z) = \min_{e \in D | e.key = z} \frac{h(z)}{e.val} = \frac{h(z)}{\mathsf{Max}_D(z)}.$$

Recall that for $X \sim \mathsf{Exp}[1]$ and $a > 0$, the distribution of $X/a$ is $\mathsf{Exp}[a]$, and that $h(z) \sim \mathsf{Exp}[1]$. Therefore, the algorithm effectively draws $\texttt{seed}_D(z) \sim \mathsf{Exp}[\mathsf{Max}_D(z)]$ for every key $z$. Moreover, from our assumption of independence of $h$, the variables $\texttt{seed}_D(z)$ of different keys $z$ are also independent.

We now notice that for a primary key $x$,

$$\texttt{seed}_D(x) = \min_{z | z.p = x} \texttt{seed}_D(z).$$

That is, $\texttt{seed}_D(x)$ is the minimum, over all keys $z$ with primary key $z.p = x$ that appeared in at least one element of $D$, of $\texttt{seed}_D(z)$.

The random variables $\mathtt{seed}_D(z)$ for input keys $z$ are independent and exponentially distributed with respective parameters $\mathsf{Max}_D(z)$. From properties of the exponential distribution, their minimum is also exponentially distributed with a parameter that is equal to the sum of their parameters $\mathsf{Max}_D(z)$:

$$\mathtt{seed}_D(x) \sim \mathsf{Exp}\left( \sum_{z | z.p = x} \mathsf{Max}_D(z) \right),$$

that is, $\mathtt{seed}_D(x) \sim \mathsf{Exp}[\mathsf{SumMax}_D(x)]$. Moreover, the independence of $\mathtt{seed}_D(x)$ (for primary keys $x$) follows from the independence of $\mathtt{seed}_D(z)$ (for input keys $z$). $\qquad\square$

## D   Proofs Deferred from Section 5

*Proof of Lemma 5.5.* Consider a fixed time during the processing of $D$ by Algorithm 4 (after some but potentially not all elements have been processed). For each key $x$, let $v_x$ be the sum of values of elements with key $x$ that have been processed so far.

For any $t > 0$, key $x \in \mathcal{X}$, and $i \in [r]$, we define an indicator random variable $I_{x,i}^t$ for the event that an element with key $(x, i)$ was generated with value less than $t$. In particular, the number of elements in Sideline is $\sum_{x \in \mathcal{X}} \sum_{i=1}^r I_{x,i}^\gamma$. The event $I_{x,i}^t = 1$ is the event that the minimum value of the elements generated with key $(x, i)$ is at most $t$. The distribution of the minimum value of these elements is $\mathsf{Exp}(v_x)$, and it follows that

$$E[I_{x,i}^t] = 1 - e^{-tv_x} \leq tv_x.$$

In particular, when $t = \gamma = \frac{2\varepsilon}{\sum_{z \in \mathcal{X}} v_z}$ and $r = \frac{k}{\varepsilon}$, we get

$$\mathsf{E}\left[ \sum_{x \in \mathcal{X}} \sum_{i=1}^r I_{x,i}^\gamma \right] \leq \sum_{x \in \mathcal{X}} \frac{r \cdot 2\varepsilon v_x}{\sum_{z \in \mathcal{X}} v_z} = 2r\varepsilon = 2k.$$

From Chernoff bounds,

$$\Pr\left[ \sum_{x \in \mathcal{X}} \sum_{i=1}^r I_{x,i}^\gamma > \left( 2 + \frac{3\ln m + 3\ln\left(\frac{1}{\delta}\right)}{2k} \right) 2k \right] \leq e^{-\frac{2}{3}k - \ln m - \ln\left(\frac{1}{\delta}\right)} \leq \frac{\delta}{m}.$$

Applying this each time an element is processed and taking union bound, we get that the size of Sideline increases beyond $4k + 3\ln m + 3\ln\left(\frac{1}{\delta}\right)$ at any time with probability at most $\delta$.

We now improve the bound to use $\log\log\left(\frac{\mathsf{Sum}_D}{\mathsf{Min}(D)}\right)$ instead of $\log m$. Let $t > 0$. Consider all the times where the value of $\gamma$ is in the interval $\left[\frac{1}{2}t, t\right]$, and for every $\gamma'$ in that interval, let $v_x(\gamma')$ denote the frequency of key $x$ at the time where $\gamma = \gamma'$. Since $\gamma$ decreases over time as elements are processed, any generated element stored in Sideline when $\gamma \in \left[\frac{1}{2}t, t\right]$ must have value at most $t$. Since only more elements are generated as $\gamma$ decreases, we can look all the elements that have been generated until $\gamma$ reached $\frac{1}{2}t$.[10]

We bound the number of elements with value at most $t$ that have been generated until the time where $\gamma = \frac{1}{2}t$. From the way we set $\gamma$ in Algorithm 4, we get as long as $\gamma \geq \frac{1}{2}t$, $\sum_{x \in \mathcal{X}} v_x(\gamma)t \leq 4\varepsilon$. Now, consider the indicator $I_{x,i}^t$ as defined above for the time where $\gamma = \frac{1}{2}t$. The number of elements stored in Sideline at any time when $\gamma \in \left[\frac{1}{2}t, t\right]$ is at most $\sum_{x \in \mathcal{X}} \sum_{i=1}^r I_{x,i}^t$. We get that

$$\mathsf{E}\left[ \sum_{x \in \mathcal{X}} \sum_{i=1}^r I_{x,i}^t \right] \leq r \sum_{x \in \mathcal{X}} t \cdot v_x(t/2) \leq 4r\varepsilon = 4k$$

and using Chernoff bounds,

$$\Pr\left[ \sum_{x \in \mathcal{X}} \sum_{i=1}^r I_{x,i}^t > \left( 2 + \frac{3\ln\left\lceil \log\left( \frac{\mathsf{Sum}_D}{\mathsf{Min}(D)} \right) \right\rceil + 3\ln\left(\frac{1}{\delta}\right)}{4k} \right) 4k \right] \leq e^{-\frac{4}{3}k - \ln\left\lceil \log\left( \frac{\mathsf{Sum}_D}{\mathsf{Min}(D)} \right)\right\rceil - \ln\left(\frac{1}{\delta}\right)}$$

$$\le \frac{\delta}{\lceil \log \left( \frac{\mathsf{Sum}_D}{\mathsf{Min}(D)} \right) \rceil}. \tag{10}$$

Finally, the minimum value $\gamma$ can get is $\frac{2\varepsilon}{\mathsf{Min}(D)}$, and the maximum value is $\frac{2\varepsilon}{\mathsf{Sum}_D}$. Hence, we can divide the interval of possible values for $\gamma$ into $\lceil \log \left( \frac{\mathsf{Sum}_D}{\mathsf{Min}(D)} \right) \rceil$ intervals of the form $\left[ \frac{1}{2}t, t \right]$, and apply the bound in Equation (10) to each one of them. By the union bound, we get that the probability that the size of Sideline exceeds $8k + 3\ln \lceil \log \left( \frac{\mathsf{Sum}_D}{\mathsf{Min}(D)} \right) \rceil + 3\ln \left( \frac{1}{\delta} \right)$ at any time during the processing of $D$ is at most $\delta$. $\qquad\square$

*Proof of Lemma 5.8.* Using Lemma 5.9, for every key $x$,

$$\begin{aligned}
\mathsf{E}[V_x] &\ge \mathsf{E}\left[ \frac{1}{r} \, \mathsf{SumMax}_E(x) \right] \\
&= \frac{1}{r} \sum_{i=1}^{r} \mathsf{E}[\mathsf{Max}_E((x,i))] \\
&= \frac{1}{r} \sum_{i=1}^{r} \int_{\gamma}^{\infty} a(t)(1 - e^{-\nu_x t})dt \qquad \text{[By Lemma 5.9]} \\
&= \int_{\gamma}^{\infty} a(t)(1 - e^{-\nu_x t})dt \\
&\ge \int_{\gamma}^{\infty} a(t)(1 - e^{-\nu_x \gamma})dt \\
&= A(\gamma)(1 - e^{-\nu_x \gamma}).
\end{aligned}$$

Recall that $\gamma = \frac{2\varepsilon}{\mathsf{Sum}_D}$. Then, using $1 - e^{-x} \ge \frac{x}{2}$ for $0 \le x \le 1$,

$$\begin{aligned}
\mathsf{E}[V] &= \mathsf{E}\left[ \sum_{x \in \mathcal{X}} V_x \right] \\
&\ge \sum_{x \in \mathcal{X}} A(\gamma)(1 - e^{-\nu_x \gamma}) \\
&= \sum_{x \in \mathcal{X}} A(\gamma) \left( 1 - e^{-\nu_x \cdot \frac{2\varepsilon}{\mathsf{Sum}_D}} \right) \\
&\ge \sum_{x \in \mathcal{X}} A(\gamma)\nu_x \cdot \frac{\varepsilon}{\mathsf{Sum}_D} \\
&= \frac{A(\gamma)\varepsilon}{\mathsf{Sum}_D} \sum_{x \in \mathcal{X}} \nu_x \\
&= A(\gamma)\varepsilon.
\end{aligned}$$

Since $r \ge \frac{k}{\varepsilon}$, we conclude that

$$\mathsf{E}[V] \ge \frac{A(\gamma)}{r} \cdot k.$$

$\qquad\square$

*Proof of Lemma 5.12.* Consider a key $x$. The seed $\mathsf{seed}^{(F)}(x)$ in the output sample is the minimum of $\mathsf{seed}^{(1)}(x)$ and $\mathsf{seed}^{(2)}$, which are the seed values obtained by the scaled PPSWOR and the SumMax samples, respectively.

The scaled PPSWOR sample is computed with respect to the weights $\nu_x B(\gamma)$, and thus $\mathsf{seed}^{(1)}(x) \sim \mathsf{Exp}[\nu_x B(\gamma)]$. Therefore using the density function of $\mathsf{Exp}[\nu_x B(\gamma)]$, we get that for all $t > 0$,

$$p_1 = \Pr[\mathsf{seed}^{(1)}(x) > t] = \exp(-\nu_x B(\gamma)t) .$$

The scaled $\mathsf{SumMax}$ sample is a PPSWOR sample with respect to weights $\frac{1}{r}\,\mathsf{SumMax}_E(x)$. Therefore, $\mathtt{seed}^{(2)}(x) \sim \mathsf{Exp}[\frac{1}{r}\,\mathsf{SumMax}_E(x)]$. Note however that $\frac{1}{r}\,\mathsf{SumMax}_E(x)$ is itself a random variable and in particular, the value $\mathsf{SumMax}_E(x)$ is not available to us with the sample. We recall that $\mathsf{SumMax}_E(x) = \sum_{i=1}^r \mathsf{Max}_E((x,i))$ where $\mathsf{Max}_E((x,i))$ are i.i.d. random variables. Using properties of the exponential distribution, we know that $\mathsf{Exp}[\frac{1}{r}\,\mathsf{SumMax}_E(x)]$ is the same distribution as the minimum of $r$ independent random variables drawn from $\mathsf{Exp}[\frac{1}{r}\,\mathsf{Max}_E((x,1))], \ldots, \mathsf{Exp}[\frac{1}{r}\,\mathsf{Max}_E((x,r))]$. Therefore, for $t > 0$,

$$\Pr[\mathtt{seed}^{(2)}(x) > t] = \prod_i \Pr\left[\mathsf{Exp}\left[\frac{1}{r}\,\mathsf{Max}_E((x,i))\right] > t\right] .$$

We now express $\Pr[\mathsf{Exp}[\frac{1}{r}\,\mathsf{Max}_E((x,i))] > t]$ using the fact that $\mathsf{Max}_E((x,i)) = A(\max\{y,\gamma\})$ for $y \sim \mathsf{Exp}[\nu_x]$:

$$\begin{aligned}
p_2 &= \Pr\left[\mathsf{Exp}\left[\frac{1}{r}\,\mathsf{Max}_E((x,i))\right] > t\right] \\
&= \int_0^\infty \nu_x \exp(-\nu_x y) \Pr\left[\mathsf{Exp}\left[\frac{1}{r}A(\max\{y,\gamma\})\right] > t\right] dy \\
&= \int_0^\infty \nu_x \exp(-\nu_x y) \exp(-A(\max\{y,\gamma\})t/r) dy .
\end{aligned}$$

Since $\Pr[\mathtt{seed}^{(2)}(x) > t] = p_2^r$ and using the fact that $\mathtt{seed}^{(1)}(x)$ and $\mathtt{seed}^{(2)}(x)$ are independent, we conclude that

$$\begin{aligned}
\Pr[\mathtt{seed}^{(F)}(x) < t] &= 1 - \Pr[\min\{\mathtt{seed}^{(1)}(x), \mathtt{seed}^{(2)}(x)\} > t] \\
&= 1 - \Pr[\mathtt{seed}^{(1)}(x) > t]\Pr[\mathtt{seed}^{(2)}(x) > t] \\
&= 1 - p_1 p_2^r .
\end{aligned}$$

$\square$