[Reviews · NeurIPS 2019]

Reviewer 1



1. The submission reads well. The introduction part is written clearly with well-turned phrases. 2. The submission demonstrates the quality of the proposed sketch by comparing a resulting variance upper bound with that for PPSWOR. I am slightly concerned about the fact that upper bounds are compared here. Often, an upper does not reflect the true scale of a quantity. It would be better if the authors can derive a certain form of lower bound for PPSWOR, and show that the induced variance bound of the proposed method is not much larger. The submission indeed mentioned, in line 93, that the bounds for PPSWOR are "tight". I would like to know how tight these bounds are, i.e., are they provably near-optimal? or just hard to be improved further? 3. Line 202 mentioned that the new sketch scheme is "guided by the sketch ... due to Cohen [9]". I am slightly concerned about the novelty of the new approach. It would be great if the authors can clarify the similarities and differences between the two sketches. 4. The supplement is an extended version of the main paper. I think it would be better to separate the detailed technical parts from the main paper and put them in the appendix. In other words, make the supplement supplementary. The current form is fine as well. 5. It is mentioned in line 100 that "all (previous methods) require aggregating the data", while line 114 states that PPSWOR also "extends to the unaggregated datasets". It would be better if the authors could explain the latter claim more clearly. 6. Maybe better to explain the significance of Section 5 somewhere in the paper. I also feel that Section 5, according to its content, should not appear between Section 4 and 6. 7. The experimental results look good, and the improvement is significant. 8. Minor: line 127 "moments with" instead of "moments wtih"; line 187 better to emphasize the phrase "complement Laplace transform" for consistency; line 219 what is "conditional inclusion probabilities" referring to?

Reviewer 2



This paper considers the problem of sampling from an unaggregated dataset in the form of key-value (x,v) pairs. Let nu_x be the frequency of the key 'x' in the dataset and let f(.) be any concave sublinear function. A typical problem considered in this paper is that of estimating \sum_{x \in H} f(nu_x), for a subset of keys H. If one had access to the aggregated data, then the optimal way is to sample key 'x' with probability proportional to f(nu_x). When f(.) is an identity function, PPSWOR is a randomized algorithm using exponential random variables that achieves this even on unaggregated datasets. PPSWOR can be considered as optimal sampling procedure in this sense for any f(.) if applied to aggregated data. This paper considers all concave sublinear f and unaggregated datasets. The full algorithm requires two passes over the data. For the first pass, authors give a composable sketching algorithm to sample keys. Once a sample of keys is generated, authors do a second pass over the data to compute the associated estimators. They show that the variance of the estimator (of f(\nu_x)) so produced is at most a constant factor worse than that of PPSWOR applied to aggregated dataset. In addition, the space required by the algorithm is of the same order as PPSWOR in expectation. Any concave sublinear function can be written as an integral over 0 to oo, and following earlier papers, the authors write this as a sum of two integrals. The authors then give two separate sampling procedures to handle each part, and give a merging procedure to combine the two samples. The first summand is approximately equal to scaled frequency of the keys, and so PPSWOR can be used to produce a sample for this part. The second summand is handled by a procedure that authors call SumMax. SumMax (Section 5) is a procedure that samples (primary part of the) keys according to a sum of max values. I find the result to be interesting and potentially useful. The techniques are interesting. One thing that was not clear to me was which part was technically the most interesting/novel contribution of this paper given [9] (and possibly other papers). One main complaint I have is that the conference version of the paper does a very poor job explaining the various parts of the algorithm. Section 3 has some explanation of where other sections fit into the overall scheme. But Section 4 and Section 5 do not fit in to the narrative and there is no/very little explanation of why these are needed for the problem. The supplementary portion is a complete paper in itself and is well written. But it was very unclear to me why stochastic version of PPSWOR (section 4) and SumMax sampling (section 5) were required before looking into the supplementary section. --Post Author Feedback--- After looking at other reviews and authors' feedback, I will keep my score unchanged (or probably raise my overall score by 0.5 points). I appreciate the clarity of the feedback, and I am satisfied with their answers.

Reviewer 3



I find the problem well-motivated. The paper is fairly well-written and provides enough context before diving into details. The techniques are also interesting and the end-result resolves an important problem. There are some loose ends such as the large constants in the approximation factor. But overall, I see this paper in the accept regime.

Reviewer 4



Review Update: The author did a better job of explaining how the PPSWROR sketch, SumMax sketch, Sideline data structure, and gamma threshold work together in Algorithm 3 in the rebuttal. ------------------------------------------------------------------------- -- Summary -- Massive datasets contain elements represented as key, value pairs. The task is to compute statistics where each key is weighted by a function of its frequency. The paper specifically focuses on concave sublinear functions to mitigate the effect of keys with high frequencies. The paper designed a composable sampling sketch for any concave sublinear function while requiring space near to the desired sample size. The stochastic PPSWOR sampling and the auxiliary SumMax sampling sketch support generating a sample of keys whose expected weight is equal to the key's true weight. Naive Approach: 1. Aggregate all data into a table of keys and their frequencies. 2. Apply function f to frequency values 3. Estimate statistics using weighted sampling scheme Probability Proportional to Size and WithOut Replacement (PPSWOR): 1. Sample keys in proportion to their frequency 2. Estimate statistics using function f * Requires space equal to the number of distinct keys

[Author Response · NeurIPS 2019]

We thank the reviewers for providing thorough and helpful reviews!

**High-level overview of components, novelty, and comparison with [9].** One concern that was raised by multiple
reviewers is that the submission does not explain how the components described in Sections 4 and 5 fit in the narrative,
and which parts are the most interesting/novel technical contributions. We provide here an overview and will update the
camera-ready version accordingly.

In Section 3, we represent $f(\nu_x) = \mathcal{L}^c[a](\nu_x)_0^\gamma + \mathcal{L}^c[a](\nu_x)_\gamma^\infty$, and for each summand we maintain a separate sample
of size $k$ (which will later be merged). This representation was used in [9] for the simpler task of estimating the full
$f$-statistics of the data but in this work we need to produce samples according to these contributions.

For $\mathcal{L}^c[a](\nu_x)_0^\gamma$, we maintain a standard PPSWOR sketch. However, to facilitate sampling according to $\mathcal{L}^c[a](\nu_x)_\gamma^\infty$, we
needed to develop novel techniques. The first challenge was to get a handle on the contributions $\mathcal{L}^c[a](\nu_x)_\gamma^\infty$, and for
that purpose we designed the SumMax sketch structure (Section 5). The input elements are mapped to output elements
with multiple subkeys, so that the sum over subkeys of the maximum element is a random variable with expectation
$\mathcal{L}^c[a](\nu_x)_\gamma^\infty$. The SumMax sampling structure provides a PPSWOR sample according to this sum of maxima.

Another challenge is that for each $x$, $\mathsf{SumMax}(x)$ is not the exact value $\mathcal{L}^c[a](\nu_x)_\gamma^\infty$, but a random variable with this
value in expectation. For that, we introduce the analysis of PPSWOR with stochastic inputs (Section 4). In that analysis,
we establish the conditions that are needed in order for the sample according to the random values to be close to a
sample according to the expected values. We then design a sketch structure that meets these conditions.

The most interesting technical contributions of the paper, that are also of independent interest, are the SumMax sampling
structure and especially the analysis of PPSWOR with stochastic inputs, which shows that under relatively mild
conditions, we can get a sample with good estimators when the input to PPSWOR is randomized.

To clarify following the questions posed by Reviewer 4: Each of the two samples we maintain (the PPSWOR and
SumMax samples) have a fixed size and store at most $k$ keys at any time. The $\gamma$ threshold is chosen to guarantee that
we get the desired approximation ratio. The only structure that can use more space is the Sideline structure. As part of
the analysis, we bound the size of the Sideline and show that in expectation, it is $O(k)$ and also provide worst case
bounds on its maximum size during the run of the algorithm. The output elements that are processed by the SumMax
sketch have a value that depends on $\gamma$ (which changes as we process the data), and the purpose of the Sideline structure
is to store elements until $\gamma$ decreases enough that their value is fixed (and then they are removed from the Sideline and
processed by the SumMax sketch).

Reviewers 1 (comment 3) and 2 asked what are the differences between this submission and the results in [9]. For any
soft concave sublinear function $f$, the sketch in [9] can estimate the statistics $\sum_{x \in \mathcal{X}} f(\nu_x)$ over the entire dataset. Our
sketch outputs a sample that can be used to estimate statistics of the form $\sum_{x \in H} L_x f(\nu_x)$ for any $H \subseteq \mathcal{X}$ and $L_x \geq 0$.

It is highly non-trivial to develop the additional components needed for a sampling structure. In particular, the sketch
in [9] gives us a way to map elements into "output" elements with the desired expected value. However, using the
framework of [9] to produce a sample required the introduction of stochastic PPSWOR sampling, the SumMax sketch,
and a modification of the way $\gamma$ is chosen.

**Tightness of the upper bounds.** Reviewer 1 comment 2: The upper bounds for PPSWOR are essentially tight for
datasets that are not very skewed (and that is why it makes sense to compare to them when designing an algorithm for
the worst case). When a few elements dominate and their weight is a large fraction of the total weight, the variance can
be lower. In particular, if we look at the variance of the benchmark PPSWOR in the experimental results, we see that
the NRMSE is at the bound $1/\sqrt{k}$ in almost all experiments.

Reviewer 3: We do not have a lower bound to match our result. Reducing the constant $4$ in the theoretical analysis
is an interesting direction for further research. Going deeper into the analysis of stochastic PPSWOR, we can hope
to improve $4$ to $2$, but at the expense of using more space. However, from a practical perspective, the error that we
get in the experiments is very close to the optimal bound of PPSWOR (without the factor $4$). The additional factor
of $(1 + 1/(e - 1))^2$ when considering non-soft concave sublinear functions is needed, as our approach relies on
approximating this family using soft concave sublinear functions.

**Answers to additional questions by Reviewer 1: 5.** When we say in line 100 that all previous methods (including
PPSWOR) require aggregation, we refer to the problem of sampling with respect to a concave sublinear function to
the frequencies. In line 114, we say that if we sample according to the frequencies themselves (without applying any
function), PPSWOR can be used on unaggregated data.
**8.** In order to compute an estimate of the target statistics, we use a conditional variant of the Horvitz-Thompson
estimator ($\nu_x / \Pr[x \text{ in sample}]$). The term "conditional inclusion probabilities" in line 219 refers to the probability in
the denominator, which we explain how to compute in the supplement/full version of the paper.

[Meta-Review · NeurIPS 2019]

There is a consensus among all four reviewers. The authors are encouraged to take into account the reviews before submitting the final version.